# Astrochronology of the Paleocene-Eocene Thermal Maximum on the Atlantic Coastal Plain

Mingsong Li [1,2] ✉, Timothy J. Bralower [2], Lee R. Kump [2], Jean M. Self-Trail[3], James C. Zachos[4], William D. Rush [4,5,6] & Marci M. Robinson [3]

The chronology of the Paleocene-Eocene Thermal Maximum (PETM, ~56 Ma) remains disputed, hampering complete understanding of the possible trigger mechanisms of this event. Here we present an astrochronology for the PETM carbon isotope excursion from Howards Tract, Maryland a paleoshelf environment, on the mid-Atlantic Coastal Plain. Statistical evaluation of variations in calcium content and magnetic susceptibility indicates astronomical forcing was involved and the PETM onset lasted about 6 kyr. The astrochronology and Earth system modeling suggest that the PETM onset occurred at an extreme in precession during a maximum in eccentricity, thus favoring high temperatures, indicating that astronomical forcing could have played a role in triggering the event. Ca content data on the paleo-shelf, along with other marine records, support the notion that a carbonate saturation overshoot followed global ocean acidification during the PETM.

The Paleocene-Eocene Thermal Maximum (PETM) was an interval of global warming that occurred ca. 56 million years ago (Ma) and was characterized by a 4–5 °C global mean surface temperature increase[1]. Estimates of the total amount of carbon released during the PETM range from ~3000 Pg to more than 13,000 Pg[2–4], which span the current assessments of remaining fossil fuel reserves[5]. The PETM is considered to have the highest carbon release rates for the past 66 million years[6], although estimates of rate are still limited by the low fidelity of records. Proposed triggers for the PETM include volcanism associated with the North Atlantic Igneous Province[4,7], dissociation of methane hydrates (e.g., ref. 8), variations in Earth's orbit that controlled massive carbon release from permafrost melting or oceanic methane hydrates[9–12], and an extraterrestrial impact[13,14]. To further complicate the matter, estimates for the onset of carbon isotope excursion (CIE) at the PETM range from several years[15,16] to thousands and even tens of thousands of years[10,12,17–20] in duration.

The PETM CIE onset is defined by a negative shift of $\delta^{13}C$. Over the past few decades, considerable effort has been made to reconstruct the chronology of the CIE using astrochronology[17–19,21], [3]He isotope measurements[22,23], and modeling experiments[6,24]. At one extreme, the CIE onset was estimated to have spanned only 13 years based on assumed annual "bedding" couplets at a paleo-shelf section on the mid-Atlantic Coastal Plain[15], an assumption contradicted by evidence for coring artefacts produced via biscuiting whereby the formation is fractured during coring and drilling mud is injected in between layers. The 13-year duration is also contradicted by evidence from foraminifer accumulation rates[25,26], and carbon cycle/climate modeling[6,27]. At the other extreme are estimates ranging up to 20 kyr as derived from deep sea cores[28,29]. These estimates, however, are complicated by slow sedimentation rates coupled with carbonate dissolution and bioturbation[12,19,30]. Independent astrochronologic studies for the basinal, shallow marine, and terrestrial sites with high sedimentation rates are few[18,21,31,32] and can be complicated due to the prevailing

[1]Key Laboratory of Orogenic Belts and Crustal Evolution, MOE, School of Earth and Space Sciences, Peking University, Beijing 100871, China. [2]Department of Geosciences, Pennsylvania State University, University Park, PA 16802, USA. [3]Florence Bascom Geoscience Center, U.S. Geological Survey, Reston, VA 20192, USA. [4]Department of Earth and Planetary Sciences, University of California Santa Cruz, Santa Cruz, CA 95064, USA. [5]Department of Earth and Planetary Sciences, Yale University, New Haven, CT 06511, USA. [6]Cooperative Institute for Research in Environmental Sciences (CIRES), University of Colorado Boulder, Boulder, CO 80309, USA. ✉e-mail: msli@pku.edu.cn

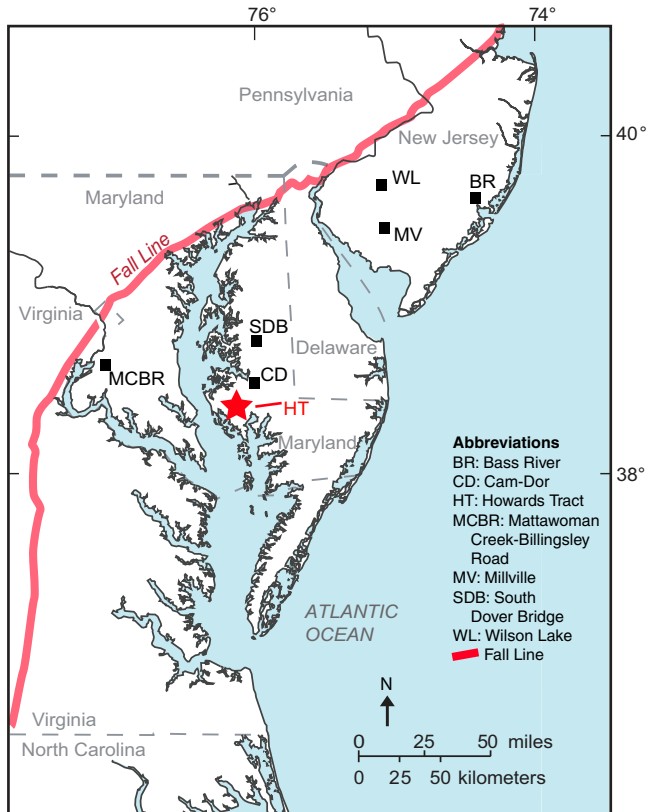

**Fig. 1 | Location of the Howards Tract cores on the mid-Atlantic Coastal Plain, the area east of the Fall Line (adapted from Bralower, T.J. et. al.[38]).** Fall line represents the paleo-shoreline of the Atlantic Ocean during the Paleocene-Eocene Thermal Maximum (PETM). BR Bass River; CD Cam-Dor; HT Howards Tract; MCBR Mattawoman Creek-Billingsley Road; MV Millville; SDB South Dover Bridge; WL Wilson Lake.

autogenetic sedimentation processes in stratigraphy[33]. Astrochronological age estimates from coastal/shelf records that have high sedimentation rates are still lacking, hindering the evaluation of the timing and the trigger of the PETM. In the coastal/shelf environment, non-orbital, $10^3$–$10^5$ year-scale sedimentary 'noise' resulting from storms, tides, bioturbation, variable sedimentation rates, short-term erosion, and diagenesis[34], as well as mobile deltaic and continental shelf muds[35,36] can be strong, hampering a straightforward interpretation of the astronomical signal in cyclostratigraphy. Moreover, although astronomical cycles have long been recognized in the PETM interval[10,18,29,37], statistical evaluation of the null hypothesis ($H_0$, no astronomical forcing) is rare, and links between astronomical forcing and proxy oscillations are unclear.

The Aquia Formation and Marlboro Clay from the Howards Tract cores (38.44827°N, 76.14159°W), two vertically offset holes in the Blackwater National Wildlife Refuge of Maryland (HT1 and HT2; Fig. 1), provide a unique opportunity to evaluate the PETM in a coastal/shelf environment using astrochronology. The Atlantic paleo-shelf sediments of the Marlboro Clay record the PETM in an exceptionally thick (5–15 m) deposit of the global low carbon value "core" of the PETM, which requires an order of magnitude faster sedimentation rate than deep-sea deposits, thus representing one of the most continuous paleo-shelf records from the mid-Atlantic Coastal Plain[38]. The spliced cores offer high temporal resolution paleoclimate proxies for the late Paleocene and early Eocene, e.g., calcium content and magnetic susceptibility (MS). Various studies demonstrate that Ca content and MS are two of the best recorders of astronomical cycles (refs. 39, 40 and references within). Ca content has long been used as a proxy of carbonate productivity in response to astronomically forced climate

change[41]. MS, a measurement of the concentration of magnetic minerals, is a proxy of detrital fluxes from terrestrial sources in the marine environment[42].

In this work, time series analysis of the proxy data (i.e., Ca content and MS) enables the recognition of astronomically forced sedimentary cycles at HT, leading to a high-resolution astrochronology for the PETM. The astrochronology is supported by statistical methods of sedimentation rate evaluation and an Earth system model of intermediate complexity. Earth system modeling of the effects of transient astronomical forcing using cGENIE provides a rare chance to elucidate the links between orbital forcing and paleoclimate proxies, e.g., Ca content, as well as the trigger to the PETM.

## Results and discussion
### Paleoclimate proxy records
The studied interval includes from base to top the glauconite-rich quartz sands of the Aquia Formation, the sandy clay to clay of the Marlboro Clay, and the clayey sand of the Nanjemoy Formation. The contact between the Aquia Formation and the Marlboro Clay is gradational with decreasing coarse fraction and $CaCO_3$ content, and a gradual color change from dark greenish gray to brownish gray. In comparison, the highly burrowed interval between the Marlboro Clay and the Nanjemoy Formation indicates a disconformable contact. The high-resolution bulk carbonate $\delta^{13}C$ record shows considerable variability at HT. Bulk carbonate $\delta^{13}C$ records indicate the PETM CIE onset spans a 60-cm-thick interval (i.e., 200.47 to 199.89 m, pink bars in Figs. 2–3), which is defined by the initial sharp decline in the $\delta^{13}C$ series and the changepoint analysis (see Methods and Supplementary Information). The magnitudes of the bulk carbonate $\delta^{13}C$ and $\delta^{18}O$ shifts at HT (Fig. 3a, b) are far larger than those from most PETM sequences, an artefact of early diagenetic carbonate siderite, common in Marlboro Clay sediments[38]. In contrast, a lower resolution benthic isotope record shows $\delta^{13}C$ and $\delta^{18}O$ shifts with magnitudes consistent with other sections along the Atlantic margin (Fig. 2b, c).

We measured Ca content using a Geotek X-ray fluorescence (XRF) scanner and magnetic susceptibility (MS) at 5 mm resolution. The XRF-generated Ca values generally match those measured in the lab (Fig. 2d), confirming the reliability of the Ca content from XRF scanning. The XRF-generated Ca content in the Aquia Formation is low (median 3.3%) with lower amplitude oscillations. The Marlboro Clay interval has a very low Ca content (median 1.9%) with occasionally more elevated values. The Ca content increases abruptly in the basal Nanjemoy Formation (median 6.3%) and remains high throughout the section (median 9.9%). MS values are relatively low in the Aquia Formation and basal Marlboro Clay and then reach their highest values in the middle Marlboro Clay, before gradually decreasing in the Nanjemoy Formation. The exceptionally low values at 182 m coincide with a minor unconformity; values rise again in the upper Nanjemoy sediments (Fig. 2f).

### Cyclostratigraphic results
Time series analysis of the proxy data below the unconformity at the base of the Nanjemoy Formation shows astronomical cycle-paced variations of the detrended $\log_{10}(Ca)$ and MS series (Fig. 4 and Supplementary Figs. 1–6). The Lomb-Scargle spectra of $\log_{10}(Ca)$ and MS show dominant wavelengths of ~12 m and 2.2–3.4 m, respectively. There are also two higher-frequency cycles at 1.0–1.2 m and 0.67–0.73 m wavelength (Fig. 4a, f). The statistical tuning of correlation coefficient (COCO) method[12] shows that optimal mean sedimentation rates are 8–15 cm/kyr, and the significance level of the null hypothesis of no orbital forcing is less than 0.05 (Fig. 4). Moreover, the average spectral misfit (ASM) method[43], which objectively evaluates potential sedimentation rates, indicates the most likely mean sediment accumulation rate is 10–16 cm/kyr (Fig. 4f, g), at which the significance level of the null hypothesis of no orbital forcing is as low as 0.0014 (Ca)

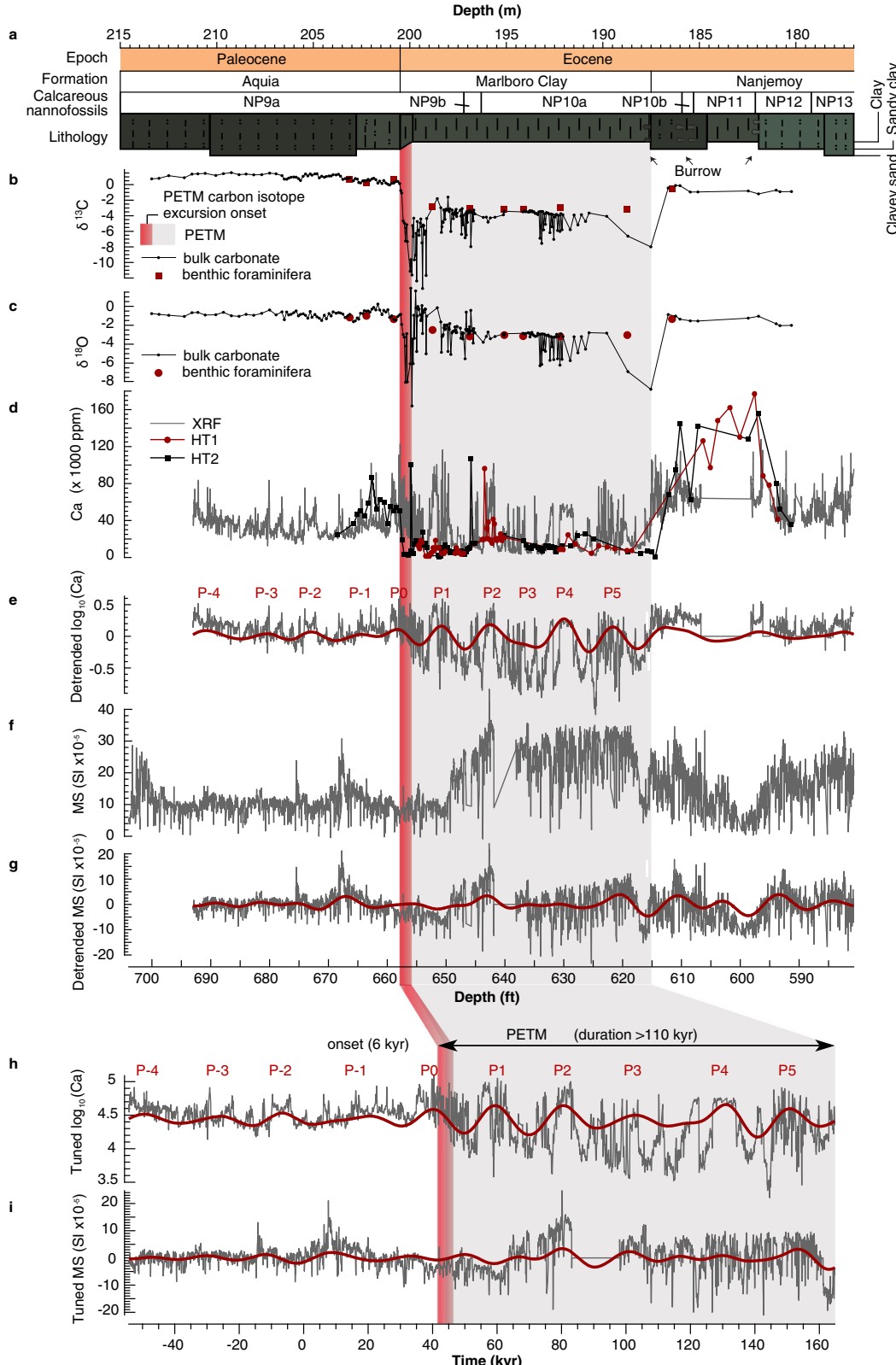

**Fig. 2 | Time series analyses. a** Chronostratigraphy, lithostratigraphy, calcareous nannofossil zones, and lithology for the Howards Tract (HT) cores. The schematic core log shows the grain size and sediment color of the cores. **b** $\delta^{13}C$ of the HT cores. **c** $\delta^{18}O$ of the HT cores. **d** Calcium content generated from X-ray fluorescence (XRF) scans of the HT cores correlates well with carbonate content from both Howards Tract 1 (HT1) and Howards Tract 2 (HT2). Ca data from 182–185 m is missing due to a coring gap. **e** Filtered 2.8 m cycles indicative of precession (P) cycles (red, Gaussian filter with a frequency of $0.35 \pm 0.15\,m^{-1}$) of the detrended $\log_{10}(Ca)$ content (gray). **f** Magnetic susceptibility (MS) of the HT cores. **g** Filtered 2.8 m cycles indicative of precession cycles (red, Gaussian filter with a frequency of $0.35 \pm 0.15\,m^{-1}$) of the detrended magnetic susceptibility. **h, i** Tuned $\log_{10}(Ca)$ (**h**) and MS (**i**) and filtered precession (P) cycles (red, Gaussian filter with a frequency of $0.05 \pm 0.018\,kyr^{-1}$). Pink bar: Paleocene-Eocene Thermal Maximum (PETM) carbon isotope excursion (CIE) onset. Shaded region: PETM.

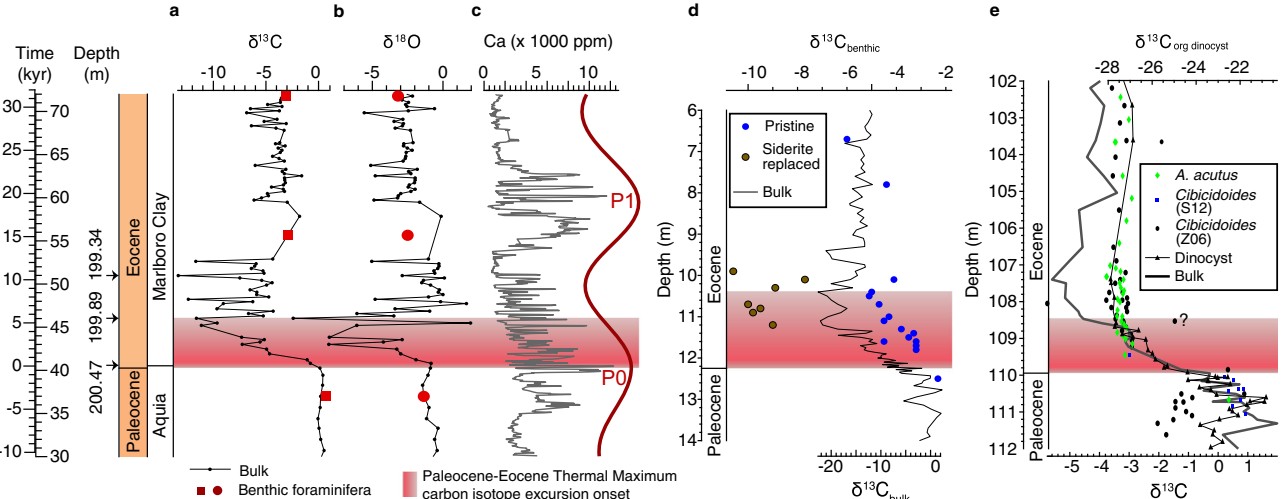

**Fig. 3 | Paleocene-Eocene Thermal Maximum (PETM) carbon isotope excursion (CIE) onset on the mid-Atlantic Coastal Plain. a** Tuned $\delta^{13}C$ values of the Howards Tract (HT) cores. **b** Tuned $\delta^{18}O$ values of the HT cores. **c** Tuned calcium content generated from X-ray fluorescence (XRF) scans (gray line) and filtered precession (P) cycles (red line, detailed in Fig. 2h) of the HT cores. **d** $\delta^{13}C$ values of the Mattawoman Creek-Billingsley Road (MCBR) cores in stratigraphic depth[20]. **e** $\delta^{13}C$ values of the Wilson Lake core in stratigraphic depth. The data source for Wilson Lake: *Cibicidoides* spp. and bulk carbonate (Z06 denotes ref. 89); *Anomalinoides acutus* and *Cibicidoides* spp. (S12 denotes ref. 53); organic matter of dinocysts[90].

and 0.0012 (MS). In other words, confidence levels of astronomically forced variations in Ca and MS are higher than 98.6%. Therefore, the ~12 m, 2.2–3.4 m, 1.0 m, and 0.63–0.75 m cycles represent ~100 kyr short eccentricity, ~20 kyr precession, and sub-Milankovitch cycles (~10 kyr and ~7 kyr), respectively.

The evolutionary fast Fourier transform (FFT), wavelet transform and Spectral Moments[44] of both Ca and MS reveals similar first-order trends in sedimentation (SI): the dominant ~2 m precession cycle at ~205–200 m increases upward gradually to an ~3 m cycle at ~185 m (Supplementary Figs. 7 and 8). This suggests the accumulation rate increased ~1.5 times from the pre-PETM to the recovery phase (Supplementary Fig. 9). Tuning of ~3 m cycles of the Ca and MS series to the 20 kyr precession cycles enables for the recognition of 5.5 precession cycles from the PETM CIE onset through the hiatus at the top of the Marlboro Clay (Supplementary Tab. 1), suggesting the Howards Tract cores preserved the lowermost 110 kyr of the PETM event (Fig. 2). Assuming the duration of each filtered precession-related cycle was 20 kyr, this astrochronology suggests the PETM CIE onset was approximately 6 kyr (Fig. 3).

### Duration of the carbon isotope excursion onset

There are two sources of uncertainty with the 6 kyr estimate for the PETM onset duration, including the definition of the CIE onset at HT and the uniformity of sedimentation rates. The Marlboro Clay is thought to have been deposited rapidly on a fluvial-deltaic-dominated shelf[36]. This energetic shelf was considered as an analog of the mobile mud belt on the modern Amazon shelf[35,45]. The combination of abundant Fe from weathering, and a suboxic early diagenetic environment in which alkalinity built up during the remineralization of organic matter via microbial sulfate reduction (cf. ref. 46), led to the precipitation of abundant siderite. The siderite formed in this early diagenetic setting incorporates low $\delta^{13}C$ from the remineralized organic matter, particularly where the primary biogenic carbonate content is low[20,38]. This would be the case during the onset, which lies within a near carbonate-free layer. As siderite formation is driven by environmental changes associated with the PETM, the global carbon isotopic excursion of ~4-5 ‰ is amplified to ~13 ‰ at HT. Moreover, because the onset of the CIE at HT coincides closely with the base of the Marlboro Clay, the possibility exists that the timing of the isotope excursion reflects both the depositional and early diagenetic

environment of the mobile mud belt as well as the input of isotopically light carbon that fueled the PETM warming recorded at sites globally.

To attempt to deconvolve these two factors, we compare the timing of the CIE onset at HT with other sections in Maryland and New Jersey where siderite is also present, yet the bulk carbonate $\delta^{13}C$ still captures the global carbon isotope signal as represented in high resolution planktonic and benthic foraminifera records from the same sections. We compare the onset with the timing of the base of the Marlboro Clay as well as three nannoplankton datums to determine whether the initial stage of the CIE at HT was more abrupt than in the other sections; such an abrupt onset could be a result of a relationship with the deposition of the mobile mud belt or early diagenetic conditions within it (Fig. 5).

Identification of datums used in this analysis can be subjective, including the base of the Marlboro Clay[38], change points in the carbon isotope excursion, and biostratigraphic datums, and we attempt to be as consistent as possible with the definition of all three types of datums (see Supplementary Information for more discussion). The analysis shows that the base of the onset of the CIE lies in an identical position to the base of the Marlboro Clay in HT as in the other two sections (Fig. 5). Moreover, the onset does not appear to be more abrupt at HT compared to Wilson Lake, New Jersey, as determined by its position relative to the three nannofossil datums, but it does appear to be two times more abrupt at HT than at South Dover Bridge, Maryland. The more abrupt onset at HT relative to the relatively close by South Dover Bridge section may be an artifact of a more condensed basal Marlboro Clay interval at HT; however, we cannot rule out the possibility that the presence of early diagenetic carbonate has made the CIE onset appear more abrupt than the original global signal. Indeed this looks to be the case at the Mattawoman Creek-Billingsley Road section in Maryland, where siderite is abundant, and the CIE onset in bulk carbonate $\delta^{13}C$ is more abrupt than in benthic foraminifera[20].

### Sedimentation rate at Howards Tract

Our estimation for the PETM CIE onset duration assumes a constant sedimentation rate within the first precession cycle (P0 in Figs. 2–3), a cycle that includes the transition between the Aquia Formation and the Marlboro Clay. The onset of the deposition of the mobile mud belt in the Marlboro Clay may have involved a significant increase in sedimentation rate that would weaken the constant sedimentation rate

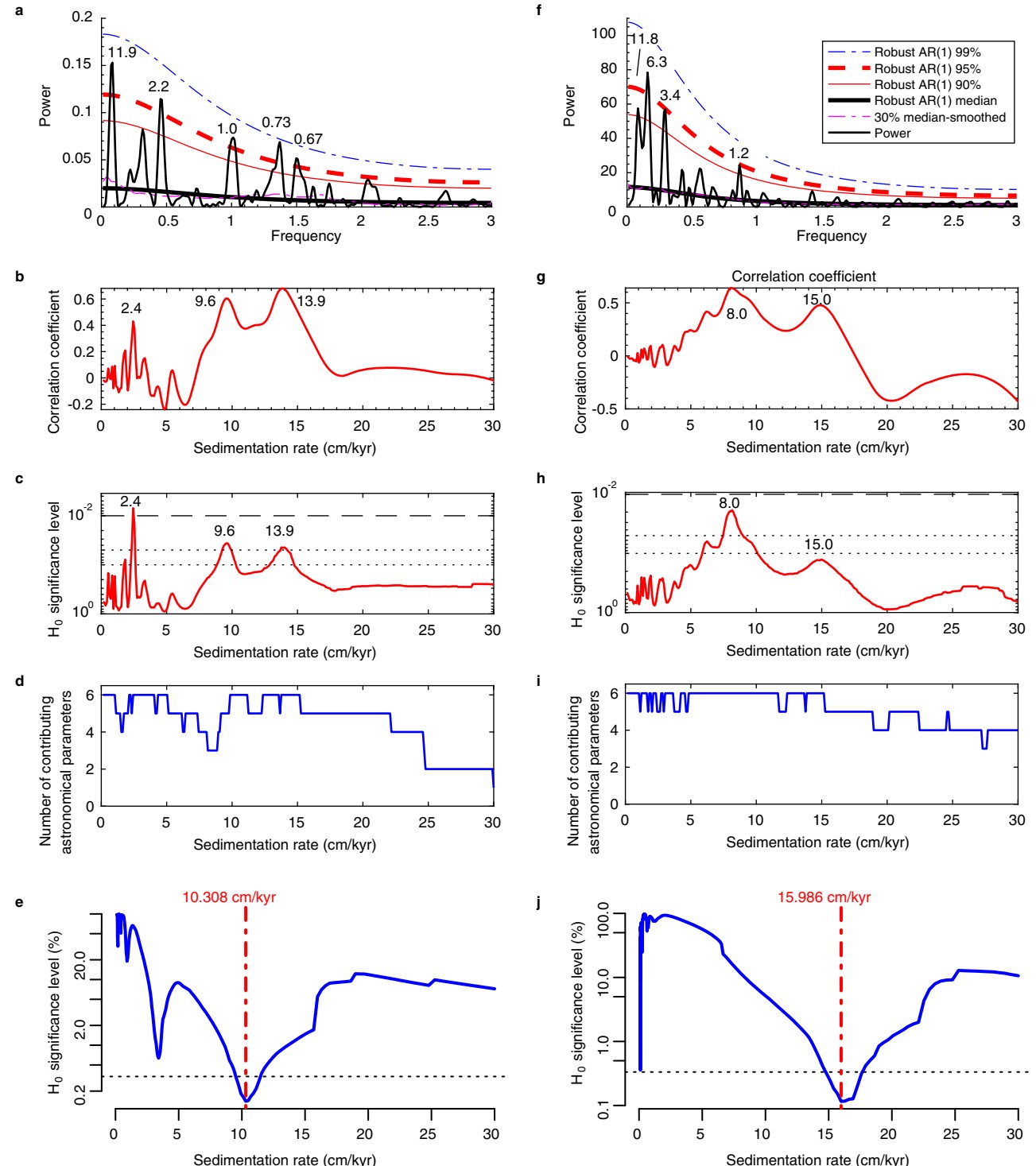

**Fig. 4 | Spectral analysis and sedimentation rate evaluation of the $\log_{10}(Ca)$ (a–e) and magnetic susceptibility (MS) (f–j) series. a, f** Lomb-Scargle spectra of the detrended $\log_{10}(Ca)$ (**a**) and detrended MS (**f**) series shown with confidence levels test against robust first-order autoregressive AR(1) red noise models. Significant periodicities are labeled with the unit of meters. The 1-slice correlation coefficient (COCO) spectra of the detrended $\log_{10}(Ca)$ (**b–d**) and detrended MS (g–i) series. Correlation coefficient (**b, g**), null hypothesis ($H_0$) significance level (**c, h**), and number of contributing astronomical parameters (See Supplementary Information for details) (**d, i**) are shown. **e, j** Average spectral misfit (ASM) of the detrended $\log_{10}(Ca)$ (**e**) and detrended MS (**j**) indicate roughly consistent mean sediment accumulation rate of 10–16 cm/kyr.

assumption, thus assuming a constant sedimentation rate would overestimate the duration of the CIE onset, which lies almost entirely in the Marlboro Clay. Nonetheless, spectral moments of both $\log_{10}(Ca)$ and MS series indicate the mean sedimentation rate within each 4 m sliding window increases only slightly between the Aquia Formation and the Marlboro Clay (Supplementary Figs. 7–9); we consider the ca.

6 kyr duration of the CIE onset determined at HT given the uncertainty related to the definition of the CIE onset and the impact of diagenesis as discussed above.

High detrital accumulation rates are thought to enhance organic carbon sequestration during the PETM, which along with the silicate weathering feedback, drove the recovery of the Earth system from

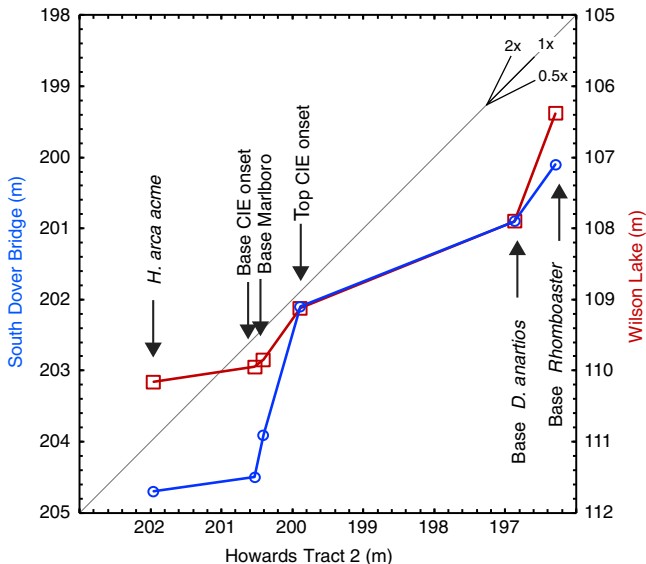

**Fig. 5 | Comparison of the carbon isotope excursion (CIE) onset with the base of the Marlboro Clay as well as three nannoplankton datums.** Blue: Howards Tract 2 versus South Dover Bridge (left); Red: Howards Tract 2 versus Wilson Lake (right). See Supplementary Information for details and discussion.

PETM $CO_2$ emissions as indicated in part by the termination of the CIE[47,48]. Indeed, many studies demonstrate that the hydrological cycle intensified during the PETM[49,50], which is reflected in large part by the dramatic increase in sedimentation rates during the PETM in continental margin settings including Belluno Basin, Italy[31], Tunisia[51], Paleotethys[52], Atlantic Coastal Plain sections[47,53], west coast of North America[47], Lomonosov Ridge in Arctic Ocean[54], and North Sea Basin[32]. However, both the compilation of hydrologic records and Earth system modeling suggests the climate response had significant regional variability – some areas are characterized by increased precipitation-evaporation, whilst others are associated with a decrease[18,21,55]. Our high-resolution astrochronology indicates that the mean sedimentation rate during the PETM at HT was ~10 cm/kyr, which is consistent with the estimates from other sites on the Atlantic Coastal Plain[53]. The evolutionary FFT, wavelet, and Spectral Moments analyses indicate a generally smooth increase in the mean sedimentation rate in the HT cores (See SI), rather than a dramatic 2.8- to 220-fold increase (i.e., from 0.1–1.0 cm/kyr for the pre-CIE to 2.8–22 cm/kyr during the CIE) in regional sedimentation rates[53]. The previous sedimentation rates were estimated via the division of the stratigraphic thickness by the corresponding duration[53], which was determined by stratigraphic correlation using biozones and the CIE shape, both of which are low resolution and can be affected by sporadic deposition and erosion in the Atlantic Coastal Plain. In comparison, the astrochronology as applied here and elsewhere is high resolution and relies on the net sediment accumulation rate.

**Astrochronology of the Paleocene-Eocene thermal maximum**
Our analysis of the Howards Tract cores is generally consistent with and more resolved than published astrochronologies and $^3$He chronological models for the PETM. Cyclostratigraphy of deep-sea cores at ODP Sites 1051 (western North Atlantic) and 690 (Weddell Sea, Southern Ocean) suggested the PETM spanned 11 precession cycles yielding a duration of 210–220 kyr, and the PETM CIE onset of initial decrease in δ$^{13}$C took over 20 kyr, while 52 kyr elapsed between the onset and the nadir of the δ$^{13}$C excursion[28,29]. About two-thirds of the excursion occurred within two steps that each was less than 1 kyr in duration, assuming a constant sedimentation rate within each precession cycle[29] (but see ref. 56). The expanded hemipelagic Forada

section (Italy) from the paleo-Tethys also records ~11 precession cycles (i.e., 231 ± 22 kyr) for the PETM[31] and the initial δ$^{13}$C decline over 12.5 cm suggesting a ~5 kyr duration based on the approximately 50 cm precession cycle[57]. Reanalysis of sedimentary records at deep sea Site 690 and sites from ODP Leg 208 (southeastern Atlantic Ocean) showed the PETM duration was ~170 kyr[19], which was supported by astrochronologic study of the Paleocene-Eocene boundary in Spitsbergen[17,58]. Cyclostratigraphy of the terrestrial Bighorn Basin site (Wyoming, USA) recognized ~7.5 precession cycles (~157 kyr) for the whole PETM[21], but a subsequent study estimated the duration of the PETM in the Bighorn Basin to be ~200 kyr[18]. Both cyclostratigraphic studies in Bighorn Basin suggested that the PETM onset occurred in less than one precession cycle[18,21]. Similarly, the astrochronology of the shallow marine Zumaia section (Spain) indicates the PETM onset lasted less than 5 kyr[48]. In comparison, assuming a constant extraterrestrial $^3$He flux, the independent $^3$He age models for the PETM suggest the duration of the whole PETM is ~120 kyr[23] or 217 kyr (+44/−33 kyr)[22]. Nonetheless, deep sea cores[29], the hemipelagic section in Italy[57], and the shallow marine section in Spain[48] are condensed, hampering a credible estimation of the onset duration. Unlike all previous estimates based on the conventional cycle-ratio approach, which can be subjective and involve circular reasoning[59], our study evaluates the null hypothesis of no orbital forcing and applies rigorous statistical tuning approaches to the chronology of the CIE onset. Here, our results suggest that the PETM record at Howards Tract spans no less than 110 kyr, though the main body of the event is truncated by an unconformity.

Our astrochronology from the same paleoshelf environment suggests that the PETM CIE onset is about 6 kyr in duration, challenging the "fast PETM onset" hypothesis[13–15] associated with the impact of a comet. Moreover, our estimate is generally consistent with those from Earth system modeling experiments that suggest the PETM CIE onset spanned at least 4 kyr[6] or less than 5 kyr[24]. The initial release of carbon at a rate of 0.6 Pg C/yr during the PETM, assuming an ~20 kyr duration of the onset, could be doubled when a 5 kyr duration is considered[4], but anthropogenic carbon release rates at ~10 Pg C/yr[60], which is one order of magnitude higher than that of the PETM. This study provides direct constraints on the carbon cycle and paleoclimate changes in the shelf environment, supporting the emerging consensus view of a few millennia for the onset interval.

**Precession forced Ca oscillations**
This study can improve our understanding of the linkage between orbital forcing and changes in paleoclimate proxies such as $CaCO_3$ content. Based on previous work using the cGENIE Earth-system model with transient orbital forcing (cf. ref. 61), we simulate the influence of transient astronomical forcing on paleoclimate to compare to our Howards Tract record. Modeling of the δ$^{13}$C excursion using cGENIE has already been undertaken[2,4,48], which forced the model to conform to observed isotope excursions, providing insightful constraints of the rate of carbon release and isotope fingerprint of the carbon source. Alternatively, we focus on astronomically forced climate change without simulating the effect of carbon release. In cGENIE model, variations of insolation are controlled by astronomical forcing[62] (Fig. 6a–c). The upper envelope of mean daily insolation at HT was paced by 20 kyr precession cycles and modulated by eccentricity cycles (Fig. 6b, d). The same is true for sea surface temperature (SST, Fig. 6e) and [$CO_3^{2-}$] ion concentration (Fig. 6f). The upper envelope of mean daily $CaCO_3$ export fluxes of biological production (Fig. 6g) or annual $CaCO_3$ fluxes (Fig. 6h) at Howards Tract is dominated by precession cycles. The annual $CaCO_3$ export fluxes compare well with the filtered precession cycles of the Ca content in the Aquia Formation and the Marlboro Clay (Fig. 6i). For example, the modeled $CaCO_3$ fluxes at ~110 kyr (i.e., 70 kyr after the PETM CIE onset) capture the minimal Ca content at 194 m (Fig. 6i).

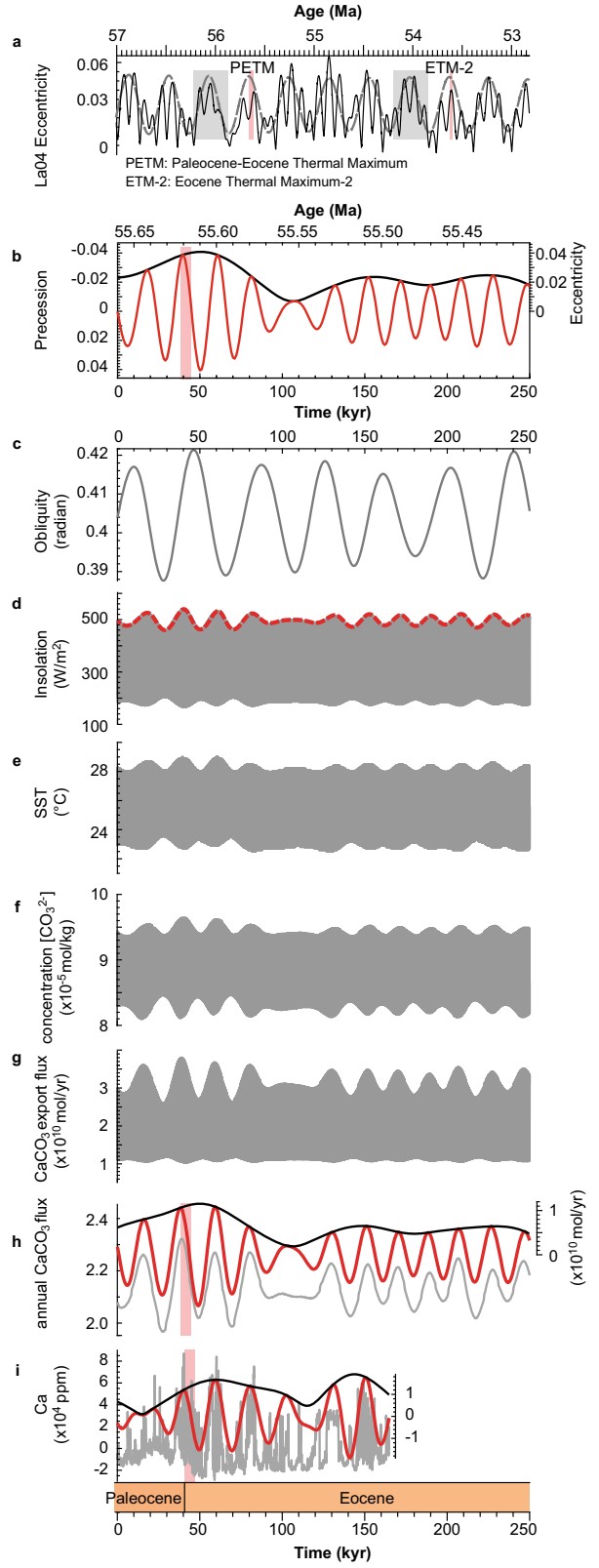

**Fig. 6 | A model-data comparison of the Ca at Howards Tract. a** Eccentricity of La2004 solution (black)[87] and filtered 405 kyr cycles (dashed gray)[12]. Gray bars in **a** denote long-term eccentricity minima following[12]. **b** Eccentricity (black) and precession (red) models of La2004 solution from 55.66 Ma to 55.41 Ma. **c** Obliquity model of La2004 solution from 55.66 Ma to 55.41 Ma. **d** Daily insolation (gray) at Howards Tract (HT) cores. Mean insolation on June 21 (dashed red) is also shown. The sampling rate is 1 kyr. **e** Daily sea surface temperature (SST) at HT cores. **f** Daily ocean surface $[CO_3^{2-}]$ concentration at HT cores. **g** Daily surface $CaCO_3$ export flux at HT cores. **h** Annual surface $CaCO_3$ export flux at HT cores (gray) shown with its Taner filtered output (red, cutoff frequency range: 0.032–0.068 kyr⁻¹) and ampli-tude modulation (black) using Hilbert transform in Acycle. **i** Tuned Ca content (gray) derived from the HT cores shown with its filtered 20 kyr cycles (red, Taner filter with a cutoff frequency of 0.032–0.068 kyr⁻¹) and amplitude modulation (black). Pink bars denote the Paleocene-Eocene Thermal Maximum carbon isotope excursion (CIE) onset (left bar in **a**, **b**, **h**, and **i**) and Eocene Thermal Maximum 2 (right bar in **a**). The frequency of daily time-series sampling is every 964 model time-steps, i.e., 20.083 year. Abbreviations: PETM, Paleocene-Eocene Thermal Maximum; ETM2, Eocene Thermal Maximum 2.

cycles and this modulation, along with variations of insolation and nutrients[67], controls the $CaCO_3$ export flux (Fig. 6f, g). Moreover, summer insolation is dominated by precession and eccentricity for-cing (Supplementary Fig. 10), while the annual insolation intensity is controlled by obliquity forcing at HT (Supplementary Fig. 11). There-fore, $CaCO_3$ flux in HT cores could be paced by astronomically forced maximum insolation, in other words, the intensity of temperature in summer season or summer half-year. A positive swing in Ca time series at HT occurs when the northern summer occurs during perihelion, and vice versa. The amplitude of Ca concentration variations is higher at eccentricity maxima because of the modulation of eccentricity, during which Earth can be either particularly close to or away from the Sun in northern hemisphere summer on the 100–405 kyr time-scale (Fig. 6h, i). Another scenario is that siliciclastic fluxes that control $CaCO_3$ content are modulated by astronomically forced changes in weathering, precipitation, runoff, and sediment discharge. In this scenario, siliciclastic dilution of $CaCO_3$ is driven partly by precipitation on the mid-Atlantic Coastal Plain which would impact sediment dis-charge and nutrient fluxes[68]. In addition, MS is considered to be a record of terrigenous material supplied to the depositional basin by runoff from the continent[42], which suggests MS should be out of phase with the Ca concentration time series. Figure 2 shows more Ca gen-erally corresponds to relatively less terrigenous material (thus drier summers), and vice versa. Therefore, the climate processes influencing the character of local sedimentation are not mutually exclusive and might enhance the lithologic cycle pattern.

## Orbital trigger for the Paleocene-Eocene thermal maximum

Forcing the cGENIE model to conform to the boron isotope pH proxy and carbon isotope data indicates a mixed source of carbon release, i.e., volcanic outgassing plus methane hydrates and/or permafrost, during the PETM onset[4,48]. Both the astrochronology in the HT cores and model results demonstrate the PETM CIE onset occurred at an extreme in precession, favoring high temperature in summers (Fig. 6b, d, e) and a maximum in the eccentricity cycles (Fig. 6a, b), indicating an astro-nomical trigger. The possibility that volcanism pulsed at the maximum in the eccentricity cycles cannot be precluded; nonetheless, increased ocean temperatures could have triggered the release of methane hydrates and/or carbon ejection from permafrost (cf. refs. 9,11). This mechanism implies that hyperthermal warming events could have occurred at other times with similar orbital configurations. Time series analysis of the deep-sea records demonstrates both the PETM and Eocene Thermal Maximum 2 (ETM-2) occurred during the eccentricity maxima[10] that post-date the very long, Myr-scale eccentricity minima (Fig. 6a)[11,12]. High-resolution paleoclimate proxy records (e.g., bulk car-bonate and benthic δ¹³C and δ¹⁸O, Fe, and $CaCO_3$) reveal that the early

The above results can be explained by the fact that insolation forcing controls surface temperature and thus determines the rate of carbonate weathering[63] and silicate weathering[64] in cGENIE[65,66] and thus the alkalinity flux to the ocean. Subsequent variations in the ocean alkalinity drive changes in the $[CO_3^{2-}]$ ion concentration in the ocean, affecting the ocean calcite and aragonite saturation states and the preservation pattern of $CaCO_3$ in the sediments (cf. ref. 61). Here maxima of $[CO_3^{2-}]$ ion concentration are modulated by precession

Eocene global warmth was punctuated by recurrent, rapid hyperthermal events, which are mainly paced by cyclicities in Earth's orbit eccentricity[69–72]. Moreover, coupled climate model simulations indicate that eccentricity-forced changes in ocean circulation and seawater temperature (through variations in seasonality) caused the destabilization of methane hydrates[73], which could explain the increasing frequency and decreasing amplitude of hyperthermal warming events in the early Eocene. Therefore, the conjunction of 100 kyr, 405 kyr, and very long, Myr-scale eccentricity cycles may have facilitated the build-up of a major mobile reservoir of reduced carbon such as methane hydrates, marine dissolved organic carbon, and/or organic-rich peat before its release during the hyperthermal events[11,70,74]. Unlike those deep ocean records that are complicated by a major dissolution interval and bioturbation in the PETM interval[10,12,19,30], high sedimentation rates and almost non-existent bioturbation in the HT cores allow for an unprecedented resolution for astrochronology of the PETM CIE onset that is supported by Earth system modeling, pointing to a possible orbital trigger for the PETM.

## CaCO₃ preservation

The HT carbonate record exhibits signs of a carbonate saturation "overshoot" in the later recovery stage of the PETM. Theory, supported by recent observations, indicates that a large and rapid release of carbon into the Earth's surface system induces a two-phase response in ocean carbonate saturation[75]. The first phase of carbon ejection will cause short-term ocean acidification lowering seawater ocean saturation ($\Omega$), while the second phase could be characterized by carbonate oversaturation caused by elevated rates of silicate weathering and elevated carbonate deposition. This phenomenon, known as carbonate saturation overshoot, could have led to an over-deepening of the calcite compensation depth (CCD) relative to its pre-event depth[75]. Globally distributed PETM sites ranging from deep ocean to shelf support ocean acidification and the shoaling of the CCD possibly to even shallow shelf depths[38]. For example, multiple cores on the Atlantic paleo-shelf in Maryland and New Jersey record an interval devoid of carbonate during the onset of the PETM and the disappearance of nannofossils and planktic foraminifera[20,38]. The dissolution of calcareous material was considered to be syndepositional possibly due to the significant shoaling of the CCD, although there are other possible explanations involving local influences, including dilution coupled with euxinia[38]. In the second phase, the recovery and overshoot in carbonate saturation is best captured in hemi-pelagic and pelagic records[30,31,76]. The Forada section in particular, with a distinct clay layer indicates resumption of carbonate deposition roughly 20 kyrs after the acidification[31]. In the Atlantic, the CCD gradually deepened over several tens of thousand of years before a state of oversaturation was reached, resulting in carbonate deposition at depths previously below the CCD[75,77]. Collectively, these observations support carbon cycle models that include a silicate weathering feedback[3,75,77]. At HT, the Nanjemoy Formation preserves a high CaCO₃ content, i.e., up to 18% in the PETM late recovery phase versus 3.3% in the pre-PETM and 1.9% during the PETM body interval (Fig. 2), demonstrating the occurrence of the overshoot in carbonate saturation. This overshoot could explain the enhanced nannofossil preservation right above the dissolution interval in cores on the mid-Atlantic paleo-shelf[38]. The trend in the Atlantic paleoshelf demonstrates the carbon saturation overshoot impacted even the shallow ocean.

To review, a statistically significant astronomical signal in the Marlboro Clay has been detected, in this case from the Howards Tract cores in Maryland. The astrochronology suggests that the Marlboro Clay at this site preserves a 110-kyr record of the PETM and that the onset of the event lasted ~6 kyr. A combination of astrochronology and Earth system modeling suggests that the PETM CIE onset occurred at an extreme in precession favoring high temperature and at the maxima

of 405 kyr and 100 kyr eccentricity cycles, indicating a possible orbital trigger. Astronomically paced siliciclastic and nutrient fluxes, along with precession-forced temperature-dependent changes in global weathering rates of carbonate and silicate rocks, could have contributed to oscillations of Ca content as exemplified at Howards Tract. Carbonate content data on the Atlantic paleo-shelf, along with other deep-sea records, suggest that carbonate saturation overshoot occurred not just in the deep sea but also in coastal regions during the PETM recovery.

## Methods

### Lithology

Two cores were drilled at Howards Tract (HT1 and HT2, 5 m apart with offset coring intervals) to minimize loss due to coring gaps. Spliced data from HT1 and HT2 resulted in relatively complete coverage for the Aquia Formation, Marlboro Clay, and Nanjemoy Formation. The Aquia Formation-Marlboro Clay contact is located at 200.43 m (657.6 ft), and the Marlboro Clay-Nanjemoy Formation contact is at 187.5 m (615.2 ft). Observation of the HT cores suggests the contact between the Aquia Formation and the overlying Marlboro Clay is very gradational. The top Aquia Formation is greenish-black, laminated sandy clay, while the overlying Marlboro Clay is laminated and silty clay with a color change gradually from dark greenish gray to brownish gray. Therefore, this is no evidence of a disconformity at the base of the Marlboro Clay at HT[38]. In comparison, a transition between the Marlboro Clay and the underlying unit has been reported at other sites, including the Medford Auger Project (MAP) cores[36], Millville[78], CamDor, and Wilson Lake[38] on the mid-Atlantic Coastal Plain. Unlike cores at Millville and Wilson Lake[15], Howards Tract cores show no evidence of couplets. Core photos are shown in Supplementary Figs. 12–15.

### Proxy data measurement

The elements in the Howards Tract cores (both HT1 and HT2) were measured using the XRF scanner of Geotek's Multi-Sensor Core Logger at Pennsylvania State University. The measurement time for calcium is 30 seconds and the spatial resolution is 5 mm. To test the reliability of the XRF-scan calcium, we measured carbonate content on a UIC Inc. coulometrics Coulometer at the University of California Santa Cruz with a precision of ± 0.05%. δ¹³C and δ¹⁸O of bulk carbonate and benthic foraminifera (3-5 specimens from the 180-212 μm size fraction of *Cibicidoides howelli* prior to the CIE and *Anomalinoides acutus* following the CIE were analyzed on a Kiel/MAT253 at the University of California Santa Cruz. Analytical precision for δ¹³C and δ¹⁸O (i.e., ±0.1‰ and ±0.16‰, respectively; 2RSD) is based on the replicate analyses of standards (i.e., Carrara Marble). All data are reported relative to Vienna Pee Dee Belemnite. The sampling rate for bulk samples is ca. 0.1 m for the top of the Aquia Formation and increases to 0.03-0.05 m for the lower part of the Marlboro Clay. Nonetheless, the base of the Marlboro Clay is characterized by a prominent interval in which CaCO₃ content decreases to close to zero during the CIE onset, i.e., the low carbonate interval (LCI) on the New Jersey and Maryland paleoshelf[38]. This LCI can be further supported by a gap in the foraminifera and very poor coccolith shield preservation, due to a lack of calcareous material during the CIE onset at many mid-Atlantic paleoshelf sites, such as South Dover Bridge[79], MCBR[20], and HT[38]. The LCI and missing cores prevent a uniformly sampling strategy and are responsible for the sampling rate over 0.3 m in the bottom of the Marlboro Clay.

### Changepoint analysis and the definition of the carbon isotope excursion onset

The changepoint analysis of δ¹³C data is able to provide the objective detection of changepoints at HT. Detailed search methods and test statistics of the changepoint analysis can be found in ref. 80. We use the cpt.meanvar function of the changepoint R package[80] because the carbon isotope data show changes in both the mean and variance. Four

changepoints are detected at depths of 186.61, 199.12, 200.47, and 202.92 m (Supplementary Text 1 and Supplementary Fig. 16). Among these, 200.47 m is used as the base of the CIE onset and coincides with the Aquia Formation-Marlboro Clay contact.

The changepoint analysis doesn't provide direct constraints for the top of the CIE onset. We choose 199.89 m (5.8 kyr after the CIE onset) as the top of the onset because this position records the largest negative $\delta^{13}C$ excursion, which is constrained by over one data point. The position at 199.34 m (11 kyr after the CIE onset) has the most negative $\delta^{13}C$ value, however, this position is only constrained by one datapoint, which is thus not used in the main paper. Even if it is used as the top of the CIE onset, the comparison of the onset with the positions for the Marlboro Clay and the three nannoplankton datums shows the onset is no more abrupt at HT compared to Wilson Lake (Supplementary Fig. 17). The results do not contradict our conclusion on the sedimentation rate variation during the CIE onset.

## Time series methods

The identification of astronomical cycles takes advantage of Acycle v2.4.1 software and follows typical procedures[58]. The Ca and magnetic susceptibility (MS) series carry a long-term trend that can be high amplitude and non-periodic, leading to power leakage from low-frequency components into the frequency band of interest[42], therefore, both series were detrended after subtracting a 20-m "loess" (local regression using weighted linear least squares and a 2nd degree polynomial model) trend for the MS series and a linear trend for the $\log_{10}(Ca)$ series. Because regularly spaced time series is required for many powerful techniques in this study, the detrended $\log_{10}(Ca)$ and MS series were interpolated using a "linear" method. To reveal the dominant wavelength of the proxy series and search for potential astronomical cycles, the Lomb-Scargle spectrum is calculated and shown with confidence levels test against robust AR(1) red noise models fitting to 30% median-smoothed power spectrum using the "Spectral Analysis" function in Acycle. Gauss and Taner bandpass filters were applied to isolate potential astronomical parameters[42]. Astronomical tuning is conducted using 20 kyr precession cycles and the "Age Scale" function in Acycle. In order to identify the sediment accumulation rate and test the null hypothesis that no astronomical forcing drove oscillations of the proxy series derived from the HT cores, we calculated the average spectral misfit (ASM)[43] using Astrochron package[81] and the correlation coefficient (COCO) spectra of the detrended $\log_{10}(Ca)$ and detrended MS series and six astronomical target periodicities (i.e., 125, 95, 39.8, 23.3, 22.0, and 18.7 kyr), which is based on the power spectrum of astronomical target series (La2004 solution from 55 Ma to 57 Ma). Details of the parameters for the ASM calculation can be found in the Supplementary Information. In the COCO calculation, classic red noise models of both spectra were removed to suppress the very high amplitude for the low frequencies. Test sedimentation rates range from 0.13 cm/kyr to 30 cm/kyr with a step of 0.1 cm/kyr. The number of Monte Carlo simulations is 2000.

TimeOpt is a statistical method for the estimation of optimal sedimentation rate for a given paleoclimate proxy series[37]. At each test sedimentation rate, the proxy series was converted from depth domain to time domain. Then TimeOpt used the Taner filter and the Hilbert transform to isolate the potential precession cycles and the corresponding amplitude envelope. The envelope was linearly regressed on a synthetic time series that was generated using eccentricity frequencies retrieved from astronomical models (e.g., La2004 solution) for a given age. The correlation coefficient of the regression at the test sedimentation rate was recorded as $r^2_{envelope}$. Meanwhile, the data were linearly regressed to another synthetic dataset using frequencies of eccentricity and precession. And the regression was reported with the correlation coefficient $r^2_{power}$. The product $r^2_{opt} = r^2_{envelope} * r^2_{power}$ was used to evaluate the most likely

sedimentation rate. The test sedimentation rate with the highest $r^2_{opt}$ might be the optimal sedimentation rate. The null hypothesis of no orbital forcing and the confidence of the optimal sedimentation rate can be evaluated using Monte Carlo simulations. The lag-1 correlation coefficient of the proxy series was calculated and used for the generation of many (e.g., 2000) red noise series. Then $r^2_{opt}$ at each test sedimentation rate of these noise series were calculated and recorded. Consequently, the percentile of the $r^2_{opt}$ using real proxy series indicates the chance of this $r^2_{opt}$ that can occur randomly. The optimal sedimentation rate can be considered to be significant when the null hypothesis of $r^2_{opt}$ is lower than 0.05. These calculations can be done either using astrochron package in R[81] or using Acycle software[58].

Here, TimeOpt analysis for the detrended $\log_{10}(Ca)$ series of the Aquia Formation and the Marlboro Clay using Acycle shows the most likely sedimentation rate is ca. 16.0 cm/kyr (Supplementary Fig. 3) at which the null hypothesis significance level of no orbital forcing is 0.001, that is, the confidence level of orbital forcing is 99.9% (Supplementary Fig. 4). The wide range of sedimentation rates at 10–17 cm/kyr demonstrates the sedimentation was variable at HT (Supplementary Fig. 4). This sedimentation rate is slightly higher than the COCO- and ASM-generated sedimentation of 8-16 cm/kyr (Fig. 4). In comparison, TimeOpt analysis for the MS series shows the most likely sedimentation rate is 8.1 cm/kyr (range of 6-10 cm/kyr; Supplementary Fig. 5) at which the null hypothesis significance level of no orbital forcing is 0.001 (Supplementary Fig. 6). This range is slightly lower than the COCO- and ASM-generated sedimentation rate of 8–16 cm/kyr. Nonetheless, all results point to the conclusion that variable 2–3 m wavelengths in our proxy records represent 20 kyr precession cycles. This is within expectation because multiple proxies and approaches can usually lead to different, but comparable within error, results[40]. Taken together, ASM, COCO and TimeOpt analyses indicate the PETM interval of the Marlboro Clay was paced by precession cycles, which were modulated by eccentricity cycles.

## Variable sedimentation rate

In order to reveal the secular trend of dominant frequencies, the evolutionary fast Fourier transform (FFT)[42] were calculated with Acycle "Evolutionary Spectral Analysis" function[58] using a sliding window of 7 m and a step of 0.01 m. Because the window size is smaller than the reported eccentricity cycles (8–10 m), precession-related cycles (2–4 m) are the strongest signal in the evolutionary FFT result. The evolutionary FFT of both Ca and MS reveals a similar trend: the dominant ~2 m precession cycles at ~205–200 m increases upward smoothly to ~3 m cycles at ~185 m (Supplementary Fig. 7), indicating an increasing upward sedimentation rate from ca. 10 cm/kyr to ~15 cm/kyr. Wavelet analysis shows cyclicities of the data series are generally stable (Supplementary Fig. 8). The ca. 11 m wavelengths that are interpreted as 100 kyr short eccentricity cycles are mostly unchanged. The 2-3 m cycles (~20 kyr precession cycles) show a similar increase upward trend.

The Spectral Moments methods evaluate the first order change in sedimentation rate via investigating the analyzed series using a periodogram with two spectral moments: mean frequency ($\mu f$) and bandwidth (B) using a sliding window approach[44]. We detect shifts and changes in sedimentation rate using the Spectral Moments method in Acycle 2.4.1[58]. The edge of the data series is fulfilled using the zero-padding method. The bandwidth and mean frequency are calculated using a sliding window of 4 m with a running step of 0.1 m. The mean sedimentation rate, required by the Spectral Moments algorithm, is set to 10 cm/kyr based on the ASM and COCO analysis. We estimate the trend in sedimentation rate by taking the LOESS trend of the bandwidth.

Spectral moments of both series are shown in Supplementary Fig. 9. The first-order changes in sedimentation rate using both Ca and MS series show an increasing upward trend. The minor discrepancies

between the two estimated sedimentation rate maps indicate complex climate responses of different proxies[40], and/or the ability of the spectral moments method in the estimation of fine-scale changes in sedimentation rate.

### Earth system modeling

We used the cGENIE Earth system model to simulate the variability of the Ca content in the HT cores. The model consists of a 3D ocean circulation model[82] coupled to a 2D energy-moisture balance model (EMBM) of the atmosphere and a dynamic-thermodynamic sea-ice model[67]. It also includes a 3D module of marine biogeochemical cycling of major nutrients, trace elements, and isotopes in the ocean[83], a 2D atmospheric chemistry module, and a module for interactions between sediments and ocean[84] and terrestrial weathering[66].

The model used a Paleogene bathymetry and continental configuration and was initialized with a value of alkalinity (1975 umol.eq.kg$^{-1}$) to produce a mean global $CaCO_3$ content of 47%[85]. Two-stage spin-up phases follow ref. 65 prior to the astronomical forcing experiment. In an initial spin-up phase, the ocean-atmosphere carbon cycle is set to 'close' with global weathering fluxes tracking sedimentary burial of $CaCO_3$ at all times and no bioturbation is allowed in the sediments[84]. This phase with a fixed $pCO_2$ value at 834 ppm and a prescribed $\delta^{13}C$ value at −4.9% lasts 20 kyr and reaches steady state at the end of the simulation. In the second phase, we set the system to 'open' to allow for the temperature-controlled carbonate and silicate weathering and $pCO_2$ free to evolve. The bioturbation is allowed for the surface sediment layer[84]. This phase lasts 200 kyr with an acceleration ratio of 1:9 (10 yr model run in every 100 yr simulation). During this experiment, the $pCO_2$ drifts within 2 ppm over 200 kyr, similar to ref. 65. In order to reproduce the astronomically forced Ca variations at HT core, the transient orbital forcing is enabled. Although some studies assign a ca. 56.0 Ma age for the PETM CIE onset[10,18], we followed[12,37,86], which indicates a ~55.6 Ma age for the PETM. Therefore, we used orbital parameters of 55.660-54.660 Ma in the La2004 solution to force cGENIE model[87]. The simulation results should not be largely affected by the choice of the onset age because both options occurred with similar orbital configurations, i.e., the peak of 405 kyr long eccentricity cycles[88]. The model was set to 'open' with bioturbation enabled and run for 300 kyr starting from 55.660 Ma, covering the entire PETM interval. The wall-clock time for one experiment is approximately 33 days.

### Data availability

The proxy series of calcium content, magnetic susceptibility, and carbon and oxygen isotopes generated in this study are provided in Supplementary Data 1.

### Code availability

The Acycle[58] used in this study are available at https://doi.org/10.5281/zenodo.3955018 and can be obtained at: https://github.com/mingsongli/acycle. Astrochron[81] can be found at https://CRAN.R-project.org/package=astrochron. The cGENIE.muffin model used is tagged as release 0.9.4 and has been assigned a DOI (https://doi.org/10.5281/zenodo.2654971). The code can be obtained at: https://github.com/derpycode/cgenie.muffin. Additional code and configuration files are provided in Supplementary Software 1.

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

## Acknowledgements

This project was funded by the National Key R&D Program of China (2021YFA0718200, M.L.) and the Heising-Simons Foundation, United States (2016–11, M.L. and L.R.K.). M.L. acknowledges the National Natural Science Foundation of China (42072040) and the Fundamental Research Funds for the Central Universities, Peking University (7100603368). T.J.B. acknowledges the National Science Foundation grant OCE-1416663. J.M.S and M.M.R. acknowledge funding from the U.S. Geological Survey Climate Research and Development Program. The experiment was done on the Domino cluster at the University of California at Riverside. Andy Ridgwell, Sandra Kirtland Turner, and Pam Vervoort are acknowledged for their help in cGENIE modeling. We thank Debra Willard and Kristin McDougall-Reid for their review of an early draft of the manuscript. Any use of trade, firm, or product names is for descriptive purposes only and does not imply endorsement by the U.S. Government.

## Author contributions

M.L. and T.J.B. designed the study, T.J.B., J.M.S, and M.M.R collected and prepared cores, M.L. conducted XRF scanning, W.D.R. collected and identified the benthic foraminifera and carried out the isotope analyses, M.L., T.J.B., L.R.K, J.M.S., and J.C.Z interpreted data. M.L. wrote the paper, and all authors contributed to editing the paper.

## Competing interests

The authors declare no competing interests.
