## [Peer Review File · Nature Communications]

Astrochronology of the Paleocene-Eocene Thermal Maximum on the Atlantic Coastal PlainREVIEWER COMMENTS

Reviewer #1 (Remarks to the Author):

Review of "Astrochronology of the Paleocene-Eocene thermal maximum on the Atlantic Coastal Plain" by Li et al.

My main comment on this manuscript is that it reports on an important new study giving the timing and duration of the PETM from paleo-shelf sediments in the mid-Atlantic area. Overall, the data look robust and support the general conclusions of the study, i.e. the "onset" of the PETM was about 5 kyr in duration and the body of the PETM recorded by these sediments was approximately 110 kyr. The authors also conclude that astronomically-forced global climate change may have been the cause of the PETM. My main criticism of the paper is that the writing is so "compressed" that some terms introduced and used by the authors need to be better explained/defined so the reader can follow the presentation of the work more easily and completely. I will give specific examples later in my review.

My scientific comments: The authors provide evidence that the onset of the PETM is coincident with an "extreme" in the Ca time series. This "extreme" is coincident with a maximum in eccentricity. The suggestion is then made that the coincidence of these two Milankovitch forcings may be a cause of the PETM. Here are some questions that come to mind that should be addressed in the manuscript. From Laskar's 2004 theoretical model of astronomical cycles, are there other times, both before and after the PETM when an "extreme" in precession coincided with a maximum in eccentricity? Is this a rare occurrence in Earth history? If there are other times when these cycles coincided, why didn't "PETMs" occur at these other times? What is an "extreme" in the Ca cycle in this study? Apparently, a positive swing in Ca time series (higher Ca concentrations) which is interpreted to mean summer occurs when the Earth is closest to the sun (maximum eccentricity means perihelion is particularly close on the 100-405 kyr time-scale. By using the term "extreme" as a description of this swing in the time series do you mean more than just a positive swing in the precession? Is it particularly high amplitude, is this the point (because of modulation by eccentricity)? I think this might be better spelled out in the text. This explanation/model is all based on the Ca time series, but I'd be curious what the MS time series is showing at this time (the onset). No mention is made of the MS data at this point in the manuscript. If the MS measurements are a record of terrigenous material supplied to the depositional basin by runoff from the continent, then I would expect MS to be out of phase with the Ca, more Ca and relatively less terrigenous material.....a quick look at the figure (2) suggests MS is out of phase with the Ca concentration time series. The implications of this correspondence should be considered in your explanation for the possible cause of the PETM. The interpretation of an "extreme" in precession-driven Ca concentration is hotter summers, but if there's less terrigenous material (based on an out of phase MS signal), it also means drier summers. Does the eGENIE model support this interpretation? Finally, the coincidence of the positive swing in precession-driven Ca with a maximum in eccentricity, is not based on the data. The data just show a coincidence between the "extreme" in the precession cycle and the "onset" of the PETM. The coincidence between that precession "extreme" cycle and a maximum in eccentricity comes from comparing the theoretical cycles calculated from Laskar 2004 and their amplitude at the onset of PETM, at about 56 Ma, not from comparing eccentricity and precession cycles observed in the same dataset (time series). This is what I infer from Figure 5 and its caption, but missed this point in the text, if it's there. It needs to be there in the text. If there is the wrong age for the PETM onset, this whole coincidence falls apart.

Some further explanation for the following is needed:

This paper makes important points about determining the duration of the "onset" of the PETM. I think it would help if you defined what exactly you mean by the "onset". From looking at your figures and reading the text, it appears it is the initial sharp decline in the C and O isotopic records. Actually, it's better defined in the C isotopic record than the O isotopic record presented in Figure 3. In lines, 91-96 you indicate that the higher amplitudes of the C and O isotopes in the HT cores are explained as due to the occurrence of siderite (Fe carbonate).....I don't follow your reasoning, please explain briefly....and the statement in line 94, "and to a lesser degree those in New Jersey". What are you saying here? It isn't clear.

In line 111, you mention that you detect sub-Milankovitch cycles of 10-7 kyr periods. Have these cycles been seen before at this time period (Paleocene/Eocene)? In line 172, what does the phrase "has only a minor increment" mean when referring to the sedimentation rate of the Marlboro Clay (what is a minor increment in a sedimentation rate)? In line 177, you are indicating what the duration of the onset means in terms of the rate of release of C to the atmosphere, and you state, " but anthropogenic carbon rates at ~ 10 Pg C/yr" What is your point here? It isn't clear. Are you saying that anthropogenic rates are higher than estimates for release rate at the onset of the PETM.....what point are you trying to make? In line 185, you indicate "rather than a dramatic 2.8 to 200 fold increase in regional sedimentation rates" indicated by what? Where did those numbers come from? Assuming what?

Lines 188-20 are apparently referring to the results of the eGENIE modeling. It wasn't clear to me from the text, but I gather that's what is being reported here. What is the difference between solar irradiance and insolation in your presentation? For those not totally immersed in this model, maybe a brief explanation/definition to allow the reader to follow your arguments. Does irradiance mean the intensity of the sunlight vs. insolation meaning the total amount of sunlight? Why is this distinction important? In line 199, what are "export fluxes" of CaCO₃? Is this the erosion/transport of Ca and carbonate from the continent with run-off, due to more intense chemical weathering? If so, please state clearly.

In line 212, you indicated that the annual insolation intensity (is this a synonym for "irradiance"?) is controlled by obliquity forcing "at this grid". What does "at this grid" mean or refer to? Is it a location or spatial scale or a time frame?

In line 240, you introduce the term, "saturation overshoot". You do explain later in this paragraph what you mean, but I find the use of this term a little confusing. What is the saturation "overshooting"? If something is saturated, it's at peak concentration, it's at 100%. If it overshoots, then is it supersaturated? Is that what you mean? Is it more an acidification/supersaturation cycle? Is that what you mean? I think this needs rewording.

Paragraph for lines 338-347: How do you explain a CaCO₃-determined sedimentation rate that is twice the MS-determined sedimentation rate? How do you explain getting two different sedimentation rates from the two different time series?

Lines 364-366: "We estimate the trend in sedimentation rate by taking the LOESS trend of the bandwidth and scaling it to the mean sedimentation rate of ~ 10 cm/kyr." I have no idea what this means, please rewrite more clearly.

In line 381, you mention you used a "Paleogene configuration" in the modeling. Of what? The paleogeography? Or something else?

Minor editing:

Line 282: insert "at" between "located" and "200.5 m"

Line 298: suggest "into the frequency band of interest" instead of the "interested frequency".....the frequency isn't interested in anything.

Line 303: here you start using "dominated" as a modifier for "wavelength". You don't mean "dominated" you mean "dominant" There are other instances in the text, please fix them all.

Lines 308-309: I would suggest replacing "potential" with "best-fit" or just eliminate it. I would also suggest "derived from the HT cores" rather than "at HT cores".

In lines 358-359, you write "invest aging". This is obviously a typo, but I don't know what you really mean. Is it "investigating"?

Ken Kodama
March 9, 2022

Reviewer #2 (Remarks to the Author):

Review: Li et al PETM Astrochronology

This paper develops an astrochronology for the PETM from a pair of cores from Howard's Tract, Maryland and deals with the duration of the PETM and most importantly its 'onset'. It also discusses the duration of the whole PETM and, to some extent, feedback mechanisms in the climate system although these different aspects of the paper are not especially well focused in the title and abstract which could be improved.

Overall I found the astrochronology relating to the body of the PETM to be well presented and argued. It is great that the authors see a statistically significant orbital signal in the Marlboro Clay which appears to be the first time that has been demonstrated. Unfortunately the hiatus at the base of the Nanjemoy Fm truncates the section so it does not provide a full estimate for PETM duration but that has been established elsewhere.

The real novelty of the paper in my estimation relates to the 'onset' of the PETM, both its position on an orbital (precession / eccentricity) maximum (already suspected from other sections but reinforced here) and its duration, which as the authors say is currently uncertain. My sense is that most of the community would put the duration at about 5 kr, based on evidence from other places reviewed in the paper, so the central result is not surprising, but it seems among the best evidence we have, so potentially a useful advance. As the authors discuss, the timing and duration of the 'onset' is important for constraining potential causes, which include possible inter-related issues such as prelude and 'precursor' events, trigger mechanisms, and carbon release mechanisms, rates and feedbacks. The evidence presented in this paper relates to all these issues and if it contributes to a better understanding of the event, it will be a significant publication.

However the paper has several weaknesses in presentation which lead me not to recommend publication until they are sorted out.

1. Basic lithological information is lacking

The first question that came to my mind is whether sedimentation is likely to have been continuous through the PETM onset at the Howard's Tract locality, because if there are any unconformities / cryptic hiatuses then cycles or parts of cycles could be missing, affecting both the timing and duration. I first looked at the 'lithology information' section of the supplemental but there are only formation names and depths given. I turned to the published literature but could not find any core descriptions or photos of Cores HT-1 and 2 – the Bralower et al. (2019) paper cited has no basic information about the cores either. I could not find anything else in the time I allowed myself for a web search and none is apparently cited in the paper, so am I right that the core description information is all unpublished? The Marlboro Clay is a formal lithologic formation defined by Glaser (1971). I also found mention of erosional unconformities at base Marlboro Clay attributed to Gibson and Bybell (1994) and also shown in other sections such as the very well-presented one on the nannofossils from South Dover Bridge by Self-Trail et al. (2012). Changes in rock formation are often unconformities because they reflect some change in sediment source or delivery. They also usually correspond to a change in sedimentation rate. That doesn't mean there is necessarily a unconformity at Howard's Tract but it does seem likely.

The only lithological information is the highly schematic core log on Fig 2 of this manuscript which only serves to confuse matters. This has a grain size scale but there are also unexplained hues which I presume are supposed to reflect sediment color – please explain. The base of the Marlboro Clay at 200.5 m does not correspond to a change in lithology; instead there is a step-change from 'clay sand' (should that be clayey sand?) to sandy clay to clay near the top of the Aquia Formation at 202.5 or 201.2 m. Rock formations are by established stratigraphic convention based on lithologic changes, so why is the base Marlboro Clay not at one of these levels? I note that base Marlboro at South Dover Bridge of Self-Trail et al. is precisely at a lithology change. Instead here it seems to have been aligned to the onset of the CIE. If so, I don't think that is 'allowable' as lithostratigraphy because it may be entirely invisible in the rock. However it may be 'good news' for continuity if the onset of the carbon isotope excursion (CIE) is not at a formation base.

To resolve all this the paper should have core photos and visual core description information as supplemental and the placement of formation boundaries must be justified within the context of the formally established lithostratigraphic definitions as set up by geologists from the Maryland Survey

and used in other local cores (I note Podrecca et al. 2021 in Geology do it differently again by recognizing a 'transition'). If there are noticeable sedimentation changes in the upper Aquia we need to know if they are just in this core or correlate elsewhere.

2. 'Onset' needs defining

'Onset' needs a definition so we know what we are talking about. Line 43 "The PETM onset is defined by a negative excursion of $\delta^{13}\text{C}$ and lasted..." is not adequate. 'Excursion' tends to mean a round-trip, out and back, and therefore the whole PETM is an excursion. The PETM is about temperature (it's in the name), not carbon isotopes as such. This distinction is not a mere quibble because there are theories out there that T change slightly preceded the carbon isotope excursion. A distinction needs to be made between how 'onset' is defined and how it is recognized in practice in this core using carbon and / or oxygen – that has implications for understanding what is being measured as a duration.

Points 1 and 2 combined meant that I found it very difficult to understand the critical interpretation paragraph lines 161-181. What are "the first couple of metres within P0" – P0 is an astronomical cycle but which two metres are being discussed that might be 'mobile mud belt diagenetic profile' and what does that actually mean, it is geological jargon I don't fully understand. Do you mean it might be very rapidly deposited sediment from river outflow? If so, doesn't that undermine the arguments about the onset timing and duration? Also the sentence 177-179 seems incomplete or ungrammatical.

Also, on line 47 it says that Self-Trail and Robinson (should be et al.) estimated the onset at ~2 kr nearby but here it is ~5 kyr, please explain discrepancy.

3. The isotope data are not adequately presented and explained

The key fact that this core contains a full record of the onset (of the CIE) seems to rely exclusively on bulk sediment isotope data (esp. carbon) but the data are presented as continuous lines making it difficult to see how well defined the 'onset' really is. I had to go to supplemental data and plot my own isotope stratigraphy graphs, but that raised questions in my mind about procedure. For most of the core the sampling seems to have been conducted at a mainly regular interval spacing of a tenth of a foot (!) but through the 'onset' the sampling density becomes irregular and there appears to be a bunch of data points missing (657.0, 657.2, 657.4, 657.6, 657.8, 648.0, 658.1, 658.3...). Please explain this irregularity. I would expect sampling frequency to be constant or even increase during critical intervals. Have data points been excluded post analysis and if so on what criteria? Is it because there was very little carbonate to analyse in the critical interval as elsewhere because of carbonate dissolution and they didn't run or had very low gas pressure (please specify and explain cut-off for acceptability). Is there a temporary absence of carbonate microfossils caused by environmental exclusion or dissolution as in other local sites? These are all interesting issues but I am currently just left wondering what the core actually shows.

Other Qs relating to the carbon isotopes:

Are the data in the Aquia sufficiently dense to rule out a 'precursor event' as has been claimed at other localities e.g, Bighorn Basin?

There are some very large magnitude $\delta^{13}\text{C}$ wiggles in the body of the CIE. The explanation for these is some kind of diagenetic overprint and the paper mentions siderite, as do other work on the coastal plain. Please outline what you mean by this, how does it actually work? How does siderite increase the apparent magnitude of the excursion? Can the influence of siderite diagenesis be picked up using the oxygen isotopes in combination?

This matters because fact that there is a reversal in the trend in mid onset which is intriguing and seen also at other adjacent sites such as Millville (Wright and Schaller) and Medford (Podrecca et al). It is way too large magnitude to be analytical error and so might be telling us something interesting. Or do you think the wiggle within the onset is also be diagenetic? Or sediment redistribution and what would be the impact of that? Similarly, why do $\delta^{13}\text{C}$ values overshoot and bounce back after the onset? – this shape to the CIE is not similar to other sections from around the world. If the very low values < -6 per mil are explained as diagenetic overprint, then one could argue that the onset is actually more like 2 kyr, bringing the petagram release rates closer to anthropogenic, which would be interesting all round.

I don't like recommending authors go back and do more work, but if there is still carbonate to be analysed, the paper would be improved by some additional and quick bulk sediment isotope analyses to at least fill in the data gaps and ideally increase resolution to see if these critical features could be better resolved. It's up to you. In any case please zoom in and somewhere plot the actual data

through the onset.

4. Presentation of the Wright / Schaller / Miller ideas.

Wright and Schaller (2014) thought they saw bedding couplets throughout the Marlboro Clay at Millville which they proposed were annual and provided a direct chronology. Myself and Ellen Thomas (2015) in *Climate of the Past* (not cited here) showed evidence from the cores themselves (concentric grooves, side injection of mud, microfaulting, etc) that the supposed couplets were a common sort of drilling artefact and therefore not a bedding feature at all. If the authors disagree with that evidence they should say why. Otherwise, line 44 should not start the discussion by stating that there are in fact bedding couplets in the Marlboro and 155-156 should not refer to bedding couplets because there aren't any.

And, as a matter of fact, does Howard's Tract show anything resembling couplets, whether sedimentary or drilling artefact?

Also, regarding 'other evidence' that the duration of the Marlboro cannot be measured in years, the microfossil accumulation rates discussed in Pearson and Thomas are also fundamental.

In summary I think all these questions are answerable and the issues can be rectified which should help the presentation.

This is the comment.

This is the response.

This is the revised text.

REVIEWER COMMENTS

Reviewer #1 (Remarks to the Author):

Review of “Astrochronology of the Paleocene-Eocene thermal maximum on the Atlantic Coastal Plain” by Li et al.

My main comment on this manuscript is that it reports on an important new study giving the timing and duration of the PETM from paleo-shelf sediments in the mid-Atlantic area. Overall, the data look robust and support the general conclusions of the study, i.e. the “onset” of the PETM was about 5 kyr in duration and the body of the PETM recorded by these sediments was approximately 110 kyr. The authors also conclude that astronomically-forced global climate change may have been the cause of the PETM. My main criticism of the paper is that the writing is so “compressed” that some terms introduced and used by the authors need to be better explained/defined so the reader can follow the presentation of the work more easily and completely. I will give specific examples later in my review.

Response: We thank Prof. Ken Kodama for the constructive comments and suggestions. We completed a major revision and presented more details about the mentioned terms.

My scientific comments: The authors provide evidence that the onset of the PETM is coincident with an “extreme” in the Ca time series. This “extreme” is coincident with a maximum in eccentricity. The suggestion is then made that the coincidence of these two Milankovitch forcings may be a cause of the PETM. Here are some questions that come to mind that should be addressed in the manuscript. From Laskar’s 2004 theoretical model of astronomical cycles, are there other times, both before and after the PETM when an “extreme” in precession coincided with a maximum in eccentricity? Is this a rare occurrence in Earth history? If there are other times when these cycles coincided, why didn’t “PETMs” occur at these other times?

Response: Actually, there were a series of rapid hyperthermal warming events in the early Eocene triggered by the release of greenhouse gases, which have been linked to eccentricity forcing. We addressed this comment by adding the following sentences in the main text:

“This mechanism implies that hyperthermal warming events could have occurred at other times with similar orbital configurations. Time series analysis of the deep-sea records demonstrates both the PETM and Eocene Thermal Maximum 2 (ETM-2) occurred during the eccentricity maxima¹ that post-date the very long, Myr-scale eccentricity minima (Fig. 6a)^{2,3}. High-resolution paleoclimate proxy records (e.g., bulk carbonate and benthic $\delta^{13}\text{C}$ and $\delta^{18}\text{O}$, Fe, and CaCO_3) reveal that the early Eocene global warmth was punctuated by recurrent, rapid hyperthermal events, which are mainly paced by cyclicities in Earth’s orbit eccentricity^{4,5,6,7}.

Moreover, coupled climate model simulations indicate that eccentricity-forced changes in ocean circulation and seawater temperature (through variations in seasonality) caused the destabilization of methane hydrates⁸, which could explain the increasing frequency and decreasing amplitude of hyperthermal warming events in the early Eocene. Therefore, the conjunction of 100 kyr, 405 kyr, and very long, Myr-scale eccentricity cycles may have facilitated the build-up of a major mobile reservoir of reduced carbon such as methane hydrates, marine dissolved organic carbon, and/or organic-rich peat before its release during the hyperthermal events^{2, 5, 9}.”

What is an “extreme” in the Ca cycle in this study? Apparently, a positive swing in Ca time series (higher Ca concentrations) which is interpreted to mean summer occurs when the Earth is closest to the sun (maximum eccentricity means perihelion is particularly close on the 100-405 kyr time-scale. By using the term “extreme” as a description of this swing in the time series do you mean more than just a positive swing in the precession? Is it particularly high amplitude, is this the point (because of modulation by eccentricity)? I think this might be better spelled out in the text.

Response: To clarify this issue, we added two sentences in the section “Precession forced Ca oscillations”:

“A positive swing in Ca time series at HT occurs when the northern summer occurs during perihelion, and vice versa. The amplitude of Ca concentration variations is higher at eccentricity maxima because of the modulation of eccentricity, during which Earth can be either particularly close to or away from the Sun in northern hemisphere summer times on the 100-405 kyr time-scale (Fig. 6h,i).”

About the note that “an extreme in the Ca cycle”, we didn’t mean the PETM onset occurred at an “extreme” in the Ca time series, alternatively, we intended to note that the PETM onset occurred “at an extreme in precession”.

This explanation/model is all based on the Ca time series, but I’d be curious what the MS time series is showing at this time (the onset). No mention is made of the MS data at this point in the manuscript. If the MS measurements are a record of terrigenous material supplied to the depositional basin by runoff from the continent, then I would expect MS to be out of phase with the Ca, more Ca and relatively less terrigenous material.....a quick look at the figure (2) suggests MS is out of phase with the Ca concentration time series. The implications of this correspondence should be considered in your explanation for the possible cause of the PETM. The interpretation of an “extreme” in precession-driven Ca concentration is hotter summers, but if there’s less terrigenous material (based on an out of phase MS signal), it also means drier summers. Does the eGENIE model support this interpretation?

Response: Although the intermediate complexity climate model cannot provide information on the precipitation and evaporation variations related to the sediment supply at HT, MS records provide constraints on the variations of dry/wet conditions. We inserted one sentence in this section:

“In addition, MS is considered to be a record of terrigenous material supplied to the depositional basin by runoff from the continent¹⁰, which suggests MS should be out of phase with the Ca concentration time series. Figure 2 shows more Ca generally corresponds to relatively less terrigenous material (thus drier summers), and vice versa. Therefore, the climate processes influencing the character of local sedimentation are not mutually exclusive and might enhance the lithologic cycle pattern.”

Finally, the coincidence of the positive swing in precession-driven Ca with a maximum in eccentricity, is not based on the data. The data just show a coincidence between the “extreme” in the precession cycle and the “onset” of the PETM. The coincidence between that precession “extreme” cycle and a maximum in eccentricity comes from comparing the theoretical cycles calculated from Laskar 2004 and their amplitude at the onset of PETM, at about 56 Ma, not from comparing eccentricity and precession cycles observed in the same dataset (time series). This is what I infer from Figure 5 and its caption, but missed this point in the text, if it’s there. It needs to be there in the text. If there is the wrong age for the PETM onset, this whole coincidence falls apart.

Response: You are right that the previous figure 5 (new figure 6) only showed a coincidence between the “extreme” in the precession cycle and the “onset” of the PETM thus this old figure didn’t support the argument on the potential trigger of the PETM. Here, we revised this figure with the eccentricity filtered output from the Ca series at HT and the Hilbert transform from the annual CaCO₃ flux from the model. Therefore, the coincidence between that precession “extreme” cycle and a maximum in eccentricity comes from both comparing the theoretical cycles calculated from Laskar 2004 and their amplitude at the onset of PETM and comparing eccentricity and precession cycles observed in the same dataset at HT.

Some further explanation for the following is needed:

This paper makes important points about determining the duration of the “onset” of the PETM. I think it would help if you defined what exactly you mean by the “onset”. From looking at your figures and reading the text, it appears it is the initial sharp decline in the C and O isotopic records. Actually, it’s better defined in the C isotopic record than the O isotopic record presented in Figure 3.

Response: We clarified these issues in the revised sentence:

“Bulk carbonate $\delta^{13}\text{C}$ records indicate the PETM CIE onset spans a 60-cm-thick interval (i.e., 200.47 to 199.89 m, pink bars in Figs. 2-3), which is defined by the initial sharp decline in the $\delta^{13}\text{C}$ series and the changepoint analysis (see Methods and Supplementary Information).”

And we have also added a new paragraph in the Methods section to explain the definition of the CIE onset at HT.

“Changepoint analysis and the definition of the CIE onset. The changepoint analysis of $\delta^{13}\text{C}$ data is able to provide the objective detection of changepoints at HT. Detailed search methods and test statistics of the changepoint analysis can be found in ref. ¹¹. We use the `cpt.meanvar` function

of the changepoint R package¹¹ because the carbon isotope data show changes in both the mean and variance. Four changepoints are detected at depths of 186.61, 199.12, 200.47, and 202.92 m (Supplementary Text 1 and Supplementary Fig. 16). Among these, 200.47 m is used as the base of the CIE onset and coincides with the Aquia Formation-Marlboro Clay contact.

The changepoint analysis doesn't provide direct constraints for the top of the CIE onset. We choose 199.89 m (5.8 kyr after the CIE onset) as the top of the onset because this position records the largest negative $\delta^{13}\text{C}$ excursion which is constrained by over one data point. The position at 199.34 m (11 kyr after the CIE onset) has the most negative $\delta^{13}\text{C}$ value, however, this position is only constrained by one datapoint, which is thus not used in the main paper. Even if it is used as the top of the CIE onset, the comparison of the onset with the positions for the Marlboro Clay and the three nannoplankton datums shows the onset is no more abrupt at HT compared to Wilson Lake (Supplementary Fig. 17). The results do not contradict our conclusion on the sedimentation rate variation during the CIE onset."

In lines, 91-96 you indicate that the higher amplitudes of the C and O isotopes in the HT cores are explained as due to the occurrence of siderite (Fe carbonate).....I don't follow your reasoning, please explain briefly....and the statement in line 94, "and to a lesser degree those in New Jersey". What are you saying here? It isn't clear.

Response: For clarification, we added more details about the reasoning:

"The magnitudes of the bulk carbonate $\delta^{13}\text{C}$ and $\delta^{18}\text{O}$ shifts at HT (Fig. 3a,b) are far larger than those from most PETM sequences, an artefact of early diagenetic carbonate siderite, common in Marlboro Clay sediments¹². In contrast, a lower resolution benthic isotope record shows $\delta^{13}\text{C}$ and $\delta^{18}\text{O}$ shifts with magnitudes consistent with other sections along the Atlantic margin (Fig. 2b,c)." And two updated paragraphs are presented in the "Discussion" section to further explain this issue.

In line 111, you mention that you detect sub-Milankovitch cycles of 10-7 kyr periods. Have these cycles been seen before at this time period (Paleocene/Eocene)?

Response: Yes. These cycles can be found at this time period from global sites. Examples include cyclostratigraphy of the Paleocene/Eocene sediments in the Bighorn Basin, Wyoming¹³, Site 690, and Site 1263.

Actually, we are going to explore these sub-Milankovitch cycles in another paper. Therefore we decided not to dig into these cycles in this paper to avoid distraction.

In line 172, what does the phrase "has only a minor increment" mean when referring to the sedimentation rate of the Marlboro Clay (what is a minor increment in a sedimentation rate)?

Response: We clarified this issue:

"Nonetheless, spectral moments of both $\log_{10}(\text{Ca})$ and MS series indicate the mean sedimentation rate within each 4-m sliding window increases only slightly between the Aquia

Formation and the Marlboro Clay (Supplementary Figs. 7-9); we consider the ca. 6 kyr duration of the CIE onset determined at HT given the uncertainty related to the definition of the CIE onset and the impact of diagenesis as discussed above.”

In line 177, you are indicating what the duration of the onset means in terms of the rate of release of C to the atmosphere, and you state, “ but anthropogenic carbon rates at ~ 10 Pg C/yr” What is your point here? It isn’t clear. Are you saying that anthropogenic rates are higher than estimates for release rate at the onset of the PETM.....what point are you trying to make?

Response: We revised the sentence:

“... anthropogenic carbon release rates at ~10 Pg C/yr¹⁴, which is one order of magnitude higher than that of the PETM”.

In line 185, you indicate “rather than a dramatic 2.8 to 200 fold increase in regional sedimentation rates” indicated by what? Where did those numbers come from? Assuming what?

Response: We completely revised this paragraph “Sedimentation rate at HT”. It introduced the assumptions and underscores the significance of the sedimentation rate estimation.

Lines 188-20 are apparently referring to the results of the eGENIE modeling. It wasn’t clear to me from the text, but I gather that’s what is being reported here. What is the difference between solar irradiance and insolation in your presentation? For those not totally immersed in this model, maybe a brief explanation/definition to allow the reader to follow your arguments. Does irradiance mean the intensity of the sunlight vs. insolation meaning the total amount of sunlight? Why is this distinction important?

Response: We added “In eGENIE model” before the introduction of the results. Moreover, the integrated solar irradiance over a given time period is called insolation. We use insolation for consistency.

In line 199, what are “export fluxes” of CaCO₃? Is this the erosion/transport of Ca and carbonate from the continent with run-off, due to more intense chemical weathering? If so, please state clearly.

Response: We clarified that the export fluxes of CaCO₃ are an output of biological production.

In line 212, you indicated that the annual insolation intensity (is this a synonym for “irradiance”?) is controlled by obliquity forcing “at this grid”. What does “at this grid” mean or refer to? Is it a location or spatial scale or a time frame?

Response: It is a spatial scale. We replace “this grid” with “HT” for clearance.

In line 240, you introduce the term, “saturation overshoot”. You do explain later in this paragraph what you mean, but I find the use of this term a little confusing. What is the saturation “overshooting”? If something is saturated, it’s at peak concentration, it’s at 100%. If it

overshoots, then is it supersaturated? Is that what you mean? Is it more an acidification/supersaturation cycle? Is that what you mean? I think this needs rewording.

Response: We rewritten these sentences:

“The HT carbonate record exhibits signs of a carbonate saturation “overshoot” in the later recovery stage of the PETM. Theory, supported by recent observations, indicates that a large fast release of carbon into the Earth’s surface system induces a two-phase response in ocean carbonate saturation¹⁵. The first phase of carbon ejection will cause short-term ocean acidification lowering seawater ocean saturation (Ω), while the second phase could be characterized by carbonate oversaturation caused by elevated rates of silicate weathering and elevated carbonate deposition. This phenomenon, known as carbonate saturation overshoot, could have led to an over-deepening of the carbonate compensation depth (CCD) relative to its pre-event depth¹⁵.”

Paragraph for lines 338-347: How do you explain a CaCO₃-determined sedimentation rate that is twice the MS-determined sedimentation rate? How do you explain getting two different sedimentation rates from the two different time series?

Response: We revised and explained this issue.

“The wide range of sedimentation rates at 10-17 cm/kyr demonstrates the sedimentation was variable at HT (Supplementary Fig. 4). This sedimentation rate is slightly higher than the COCO- and ASM-generated sedimentation of 8-16 cm/kyr (Fig. 4). In comparison, TimeOpt analysis for the MS series shows the most likely sedimentation rate is 8.1 cm/kyr (range of 6-10 cm/kyr; Supplementary Fig. 5) at which the null hypothesis significance level of no orbital forcing is 0.001 (Supplementary Fig. 6). This range is slightly lower than the COCO- and ASM-generated sedimentation rate of 8-16 cm/kyr. Nonetheless, all results point to the conclusion that variable 2-3 m wavelengths in our proxy records represent 20 kyr precession cycles. This is within expectation because multiple proxies and approaches can usually lead to different, but comparable within error, results¹⁶.”

Lines 364-366: “We estimate the trend in sedimentation rate by taking the LOESS trend of the bandwidth and scaling it to the mean sedimentation rate of ~ 10 cm/kyr.” I have no idea what this means, please rewrite more clearly.

Response: We rewrote this sentence:

“The mean sedimentation rate, required by the Spectral Moments algorithm, is set to 10 cm/kyr based on the ASM and COCO analysis. We estimate the trend in sedimentation rate by taking the LOESS trend of the bandwidth.”

In line 381, you mention you used a “Paleogene configuration” in the modeling. Of what? The paleogeography? Or something else?

Response: We added “bathymetry and continental” before the “configuration”.

Minor editing:

Line 282: insert “at” between “located” and “200.5 m”

Response: Accepted.

Line 298: suggest “into the frequency band of interest” instead of the “interested frequency”.....the frequency isn’t interested in anything.

Response: Accepted.

Line 303: here you start using “dominated” as a modifier for “wavelength”. You don’t mean “dominated” you mean “dominant” There are other instances in the text, please fix them all.

Response: All fixed.

Lines 308-309: I would suggest replacing “potential” with “best-fit” or just eliminate it. I would also suggest “derived from the HT cores” rather than “at HT cores”.

Response: Accepted.

In lines 358-359, you write “invest aging”. This is obviously a typo, but I don’t know what you really mean. Is it “investigating”?

Response: It is “investigating”. Fixed.

Ken Kodama
March 9, 2022

Reviewer #2 (Remarks to the Author):

Review: Li et al PETM Astrochronology

This paper develops an astrochronology for the PETM from a pair of cores from Howard's Tract, Maryland and deals with the duration of the PETM and most importantly its 'onset'. It also discusses the duration of the whole PETM and, to some extent, feedback mechanisms in the climate system although these different aspects of the paper are not especially well focused in the title and abstract which could be improved.

Overall I found the astrochronology relating to the body of the PETM to be well presented and argued. It is great that the authors see a statistically significant orbital signal in the Marlboro Clay which appears to be the first time that has been demonstrated. Unfortunately the hiatus at the base of the Nanjemoy Fm truncates the section so it does not provide a full estimate for PETM duration but that has been established elsewhere.

The real novelty of the paper in my estimation relates to the 'onset' of the PETM, both its position on an orbital (precession / eccentricity) maximum (already suspected from other sections but reinforced here) and its duration, which as the authors say is currently uncertain. My sense is that most of the community would put the duration at about 5 kr, based on evidence from other places reviewed in the paper, so the central result is not surprising, but it seems among the best evidence we have, so potentially a useful advance.

As the authors discuss, the timing and duration of the 'onset' is important for constraining potential causes, which include possible inter-related issues such as prelude and 'precursor' events, trigger mechanisms, and carbon release mechanisms, rates and feedbacks. The evidence presented in this paper relates to all these issues and if it contributes to a better understanding of the event, it will be a significant publication.

Response: We appreciate reviewer #2 for the constructive comments. In the revised manuscript, we highlighted the importance of the application of statistical approaches in astrochronology for the PETM. All previous estimations for the PETM onset duration were based on the conventional tuning approach, which assumed that the specific stratigraphic wavelength is the signal of precession cycles based on cycle ratios (short eccentricity: precession = 1: 5). Their assumptions have never been rigorously evaluated and thus the estimates could be very subjective. In comparison, our study uses three advanced techniques (COCO, ASM and TimeOpt) to reject the null hypothesis of no orbital forcing and provide strong evidence that the assumed precession-related stratigraphic variation can be used for astronomical tuning.

Considering that the journal has strict limits on the length of the abstract, we have no space to added more information in the abstract and elect to keep the original title.

However the paper has several weaknesses in presentation which lead me not to recommend publication until they are sorted out.

1. Basic lithological information is lacking

The first question that came to my mind is whether sedimentation is likely to have been continuous through the PETM onset at the Howard's Tract locality, because if there are any disconformities / cryptic hiatuses then cycles or parts of cycles could be missing, affecting both the timing and duration. I first looked at the 'lithology information' section of the supplemental but there are only formation names and depths given. I turned to the published literature but could not find any core descriptions or photos of Cores HT-1 and 2 – the Bralower et al. (2019) paper cited has no basic information about the cores either. I could not find anything else in the time I allowed myself for a web search and none is apparently cited in the paper, so am I right that the core description information is all unpublished?

The Marlboro Clay is a formal lithologic formation defined by Glaser (1971). I also found mention of erosional disconformities at base Marlboro Clay attributed to Gibson and Bybell (1994) and also shown in other sections such as the very well-presented one on the nannofossils from South Dover Bridge by Self-Trail et al. (2012). Changes in rock formation are often disconformities because they reflect some change in sediment source or delivery. They also usually correspond to a change in sedimentation rate. That doesn't mean there is necessarily a disconformity at Howard's Tract but it does seem likely.

The only lithological information is the highly schematic core log on Fig 2 of this manuscript which only serves to confuse matters. This has a grain size scale but there are also unexplained hues which I presume are supposed to reflect sediment color – please explain. The base of the Marlboro Clay at 200.5 m does not correspond to a change in lithology; instead there is a step-change from 'clay sand' (should that be clayey sand?) to sandy clay to clay near the top of the Aquia Formation at 202.5 or 201.2 m. Rock formations are by established stratigraphic convention based on lithologic changes, so why is the base Marlboro Clay not at one of these levels? I note that base Marlboro at South Dover Bridge of Self-Trail et al. is precisely at a lithology change. Instead here it seems to have been aligned to the onset of the CIE. If so, I don't think that is 'allowable' as lithostratigraphy because it may be entirely invisible in the rock. However it may be 'good news' for continuity if the onset of the carbon isotope excursion (CIE) is not at a formation base.

To resolve all this the paper should have core photos and visual core description information as supplemental and the placement of formation boundaries must be justified within the context of the formally established lithostratigraphic definitions as set up by geologists from the Maryland Survey and used in other local cores (I note Podrecca et al. 2021 in *Geology* do it differently again by recognizing a 'transition'). If there are noticeable sedimentation changes in the upper Aquia we need to know if they are just in this core or correlate elsewhere.

Response: The detailed core description information is not published elsewhere. We present more introduction to the core information in the Method and Supplement Information including the lithology, visual core description, and photos of the HT cores. In the "Results" section, we added

“The contact between the Aquia Formation and the Marlboro Clay is gradational with decreasing coarse fraction and CaCO₃ content, and a gradual color change from dark greenish gray to brownish gray. In comparison, the highly burrowed interval between the Marlboro Clay and the Nanjemoy Formation indicates a disconformable contact.”

And at the beginning of the method section, we added

“Observation of the HT cores suggests the contact between the Aquia Formation and the overlying Marlboro Clay is very gradational. The top Aquia Formation is greenish-black, laminated sandy clay, while the overlying Marlboro Clay is laminated and silty clay with a color change gradually from dark greenish gray to brownish gray. Therefore, this is no evidence of a disconformity at the base of the Marlboro Clay at HT¹². In comparison, a transition between the Marlboro Clay and the underlying unit has been reported at other sites including the Medford Auger Project (MAP) cores¹⁷, Millville¹⁸, CamDor, and Wilson Lake¹² on the mid-Atlantic Coastal Plain. Unlike cores at Millville and Wilson Lake¹⁹, Howards Tract cores show no evidence of couplets. Core photos are shown in Supplementary Figs. 12-15.”

Moreover, we explained the hues and grain size in the updated caption of figure 2. The “clay sand” has been revised to “clayey sand” in figure 2. The lithology in previous figure 2 is highly schematic, thus doesn’t provide clear information on the division of the rock formations. We updated the lithology of HT cores. Detailed lithology of HT cores can be found in the Supplement Information.

2. ‘Onset’ needs defining

‘Onset’ needs a definition so we know what we are talking about. Line 43 “The PETM onset is defined by a negative excursion of $\delta^{13}\text{C}$ and lasted...” is not adequate. ‘Excursion’ tends to mean a round-trip, out and back, and therefore the whole PETM is an excursion. The PETM is about temperature (it’s in the name), not carbon isotopes as such. This distinction is not a mere quibble because there are theories out there that T change slightly preceded the carbon isotope excursion. A distinction needs to be made between how ‘onset’ is defined and how it is recognized in practice in this core using carbon and / or oxygen – that has implications for understanding what is being measured as a duration.

Response: We changed the “PETM onset” to “PETM CIE onset” and clarified these issues in the revised sentence:

“Bulk carbonate $\delta^{13}\text{C}$ records indicate the PETM CIE onset spans a 60-cm-thick interval (i.e., 200.47 to 199.89 m, pink bars in Figs. 2-3), which is defined by the initial sharp decline in the $\delta^{13}\text{C}$ series and the changepoint analysis (see Methods and Supplementary Information).”

And we have also added a new paragraph in the Methods section to explain the definition of the CIE onset at HT.

“Changepoint analysis and the definition of the CIE onset. The changepoint analysis of $\delta^{13}\text{C}$ data is able to provide the objective detection of changepoints at HT. Detailed search methods and

test statistics of the changepoint analysis can be found in ref. ¹¹. We use the `cpt.meanvar` function of the changepoint R package ¹¹ because the carbon isotope data show changes in both the mean and variance. Four changepoints are detected at depths of 186.61, 199.12, 200.47, and 202.92 m (Supplementary Text 1 and Supplementary Fig. 16). Among these, 200.47 m is used as the base of the CIE onset and coincides with the Aquia Formation-Marlboro Clay contact.

The changepoint analysis doesn't provide direct constraints for the top of the CIE onset. We choose 199.89 m (5.8 kyr after the CIE onset) as the top of the onset because this position records the largest negative $\delta^{13}\text{C}$ excursion which is constrained by over one data point. The position at 199.34 m (11 kyr after the CIE onset) has the most negative $\delta^{13}\text{C}$ value, however, this position is only constrained by one datapoint, which is thus not used in the main paper. Even if it is used as the top of the CIE onset, the comparison of the onset with the positions for the Marlboro Clay and the three nannoplankton datums shows the onset is no more abrupt at HT compared to Wilson Lake (Supplementary Fig. 17). The results do not contradict our conclusion on the sedimentation rate variation during the CIE onset."

Points 1 and 2 combined meant that I found it very difficult to understand the critical interpretation paragraph lines 161-181. What are "the first couple of metres within P0" – P0 is an astronomical cycle but which two metres are being discussed that might be 'mobile mud belt diagenetic profile' and what does that actually mean, it is geological jargon I don't fully understand. Do you mean it might be very rapidly deposited sediment from river outflow? If so, doesn't that undermine the arguments about the onset timing and duration? Also the sentence 177-179 seems incomplete or ungrammatical.

Response: We have completely rewritten related sections. The updated sentences explain the term "mobile mud belt diagenetic profile" and the reason why we need to discuss it. Moreover, the cited study emphasized that this possibility doesn't alter the timing of the PETM at HT as detailed in the revised text:

"There are two sources of uncertainty with the 6 kyr estimate for the PETM onset duration including the definition of the CIE onset at HT and the uniformity of sedimentation rate. The Marlboro Clay is thought to have been deposited rapidly on a fluvial-deltaic-dominated shelf⁷. This energetic shelf was considered as an analog of the mobile mud belt on the modern Amazon shelf^{20, 21}. The combination of abundant Fe from weathering, and a suboxic early diagenetic environment in which alkalinity built up during the remineralization of organic matter via microbial sulfate reduction (cf. ref. ²²), led to the precipitation of abundant siderite. The siderite formed in this early diagenetic setting incorporates low $\delta^{13}\text{C}$ from the remineralized organic matter, particularly where the primary biogenic carbonate content is low ^{12, 23}. This would be the case during the onset which lies within a near carbonate free layer. As siderite formation is driven by environmental changes associated with the PETM, the global carbon isotopic excursion of ~4-5 ‰ is amplified to ~13 ‰ at HT. Moreover, because the onset of the CIE at HT coincides closely with the base of the Marlboro Clay, the possibility exists that the timing of the isotope excursion reflects both the depositional and early diagenetic environment of the mobile mud belt as well as the input of isotopically light carbon that fueled the PETM warming recorded at sites globally.

To attempt to deconvolve these two factors, we compare the timing of the CIE onset at HT with other sections in Maryland and New Jersey where siderite is also present, yet the bulk carbonate $\delta^{13}\text{C}$ still capture the global carbon isotope signal as represented in high resolution planktonic and benthic foraminifera records from the same sections. We compare the onset with the timing of the base of the Marlboro Clay as well as three nannoplankton datums to determine whether the initial stage of the CIE at HT was more abrupt than in the other sections; such an abrupt onset could be a result of a relationship with the deposition of the mobile mud belt or early diagenetic conditions within it (Fig. 5).

Identification of datums used in this analysis can be subjective, including the base of the Marlboro Clay¹², change points in the carbon isotope excursion, and biostratigraphic datums, and we attempt to be as consistent as possible with the definition of all three types of datums (see Supplementary Information for more discussion). The analysis shows that the base of the onset of the CIE lies in an identical position to the base of the Marlboro Clay in HT as in the other two sections (Fig. 5). Moreover, the onset does not appear to be more abrupt at HT compared to Wilson Lake, New Jersey, as determined by its position relative to the three nannofossil datums, but it does appear to be two times more abrupt at HT than at South Dover Bridge, Maryland. The more abrupt onset at HT relative to the relatively close by South Dover Bridge section may be an artifact of a more condensed basal Marlboro Clay interval at HT; however, we cannot rule out the possibility that the presence of early diagenetic carbonate has made the CIE onset appear more abrupt than the original global signal. Indeed this looks to be the case at the Mattawoman Creek-Billingsley Road section in Maryland where siderite is abundant and the CIE onset in bulk carbonate $\delta^{13}\text{C}$ is more abrupt than in benthic foraminifera²³.”

Also, on line 47 it says that Self-Trail and Robinson (should be et al.) estimated the onset at ~2 kr nearby but here it is ~5 kyr, please explain discrepancy.

Response: In that paper, the 2 kyr duration is estimated for a foraminiferal assemblage transition from cooler, oxygenated mixed-layer conditions to warm, dysoxic, stratified conditions at Mattawoman Creek-Billingsley Road (MCBR) in Maryland when a 50 cm/kyr constant sedimentation rate is assumed. And the sedimentation rate is calculated based on the assumption that the CIE onset (2 m in thickness at MCBR) has a minimum duration of 4 kyr based on the carbon cycle-climate modeling by Zeebe et al., (2016)²⁴. Therefore, we elect to remove this sentence.

3. The isotope data are not adequately presented and explained

The key fact that this core contains a full record of the onset (of the CIE) seems to rely exclusively on bulk sediment isotope data (esp. carbon) but the data are presented as continuous lines making it difficult to see how well defined the ‘onset’ really is. I had to go to supplemental data and plot my own isotope stratigraphy graphs, but that raised questions in my mind about procedure. For most of the core the sampling seems to have been conducted at a mainly regular interval spacing of a tenth of a foot (!) but through the ‘onset’ the sampling density becomes irregular and there appears to be a bunch of data points missing (657.0, 657.2, 657.4, 657.6, 657.8, 648.0, 658.1, 658.3...). Please explain this irregularity. I would expect sampling

frequency to be constant or even increase during critical intervals. Have data points been excluded post analysis and if so on what criteria? Is it because there was very little carbonate to analyse in the critical interval as elsewhere because of carbonate dissolution and they didn't run or had very low gas pressure (please specify and explain cut-off for acceptability). Is there a temporary absence of carbonate microfossils caused by environmental exclusion or dissolution as in other local sites? These are all interesting issues but I am currently just left wondering what the core actually shows.

Response: We replotted isotope data and updated figures 2 and 3. We added the details of the isotope data in the Method:

“The sampling rate for bulk samples is ca. 0.1 m for the top of the Aquia Formation and increases to 0.03-0.05 m for the lower part of the Marlboro Clay. Nonetheless, the base of the Marlboro Clay is characterized by a prominent interval in which CaCO₃ content decreases to close to zero during the CIE onset, i.e., the low carbonate interval (LCI) on the New Jersey and Maryland paleoshelf¹². This LCI can be further supported by a gap in the foraminifera and very poor coccolith shield preservation, due to a lack of calcareous material during the CIE onset at many mid-Atlantic paleoshelf sites, such as South Dover Bridge²⁵, MCBR²³, and HT¹². The LCI and missing cores prevent a uniformly sampling strategy and are responsible for the sampling rate over 0.3 m in the bottom of the Marlboro Clay.”

Other Qs relating to the carbon isotopes:

Are the data in the Aquia sufficiently dense to rule out a ‘precursor event’ as has been claimed at other localities e.g, Bighorn Basin?

Response: The precursor event (POE) is very interesting. But it is pity that we don't think our bulk carbonate $\delta^{13}\text{C}$ alone is able to provide constraints for the POE. In looking at the full $\delta^{13}\text{C}$ record in Figure 2, the fluctuation would be much smaller than any POE previously identified. No one has defined what a POE has to look like, but we don't have any SST or pH data to support that they could be a POE similar to the one at South Dover Bridge. We went back and looked at foram counts. We don't see any evidence of dissolution in the Aquia Fm or any planktic excursion taxa like we see elsewhere in the Aquia above the POE. Therefore, it might be premature to call this a POE or give it a duration estimate.

There are some very large magnitude $\delta^{13}\text{C}$ wiggles in the body of the CIE. The explanation for these is some kind of diagenetic overprint and the paper mentions siderite, as do other work on the coastal plain. Please outline what you mean by this, how does it actually work? How does siderite increase the apparent magnitude of the excursion? Can the influence of siderite diagenesis be picked up using the oxygen isotopes in combination?

This matters because fact that there is a reversal in the trend in mid onset which is intriguing and seen also at other adjacent sites such as Millville (Wright and Schaller) and Medford (Podrecca et al). It is way too large magnitude to be analytical error and so might be telling us something interesting. Or do you think the wiggle within the onset is also be diagenetic? Or sediment redistribution and what would be the impact of that? Similarly, why do $\delta^{13}\text{C}$ values overshoot

and bounce back after the onset? – this shape to the CIE is not similar to other sections from around the world. If the very low values < -6 per mil are explained as diagenetic overprint, then one could argue that the onset is actually more like 2 kyr, bringing the petagram release rates closer to anthropogenic, which would be interesting all round.

I don't like recommending authors go back and do more work, but if there is still carbonate to be analysed, the paper would be improved by some additional and quick bulk sediment isotope analyses to at least fill in the data gaps and ideally increase resolution to see if these critical features could be better resolved. It's up to you. In any case please zoom in and somewhere plot the actual data through the onset.

Response: We expanded the discussion on the diagenesis and siderite. The updated text can be found above and not repeated here. We zoomed in and plotted the actual isotope data in the updated figure 3.

For the $\delta^{13}\text{C}$ wiggle within the PETM CIE onset, the detailed structure of CIE onset is critical for understanding the trigger mechanisms and consequences of the PETM, however, the “wiggle” (the break in decreasing $\delta^{13}\text{C}$ values) within the CIE onset at HT might be caused by the presence/absence of siderite. Therefore, we don't have evidence to discuss this “event” at HT in this paper.

The compilation shows the bulk $\delta^{13}\text{C}$ “wiggle” is globally distributed although there may be multiple causes including more global factors such as the turnover of nannoplankton and local ones such as the abundance of siderite at HT. The “wiggle” within the onset can be observed at global marine sites from high-resolution $\delta^{13}\text{C}$ records of bulk carbonate^{17, 19, 23, 26, 27, 28, 29, 30, 31, 32} and organic matter³³ demonstrating the “wiggle” could be a fingerprint of a global event (figure below). The CIE and the onset “wiggle” have been interpreted as multiple methane releases^{27, 34}. However, the “wiggle” is not seen in any foraminiferal $\delta^{13}\text{C}$ records within the CIE onset even in high-resolution $\delta^{13}\text{C}$ records at Site 690³⁵ and MCBR²³ possibly due to winnowing at the former site and the paucity of specimens at the latter site. Changes in bulk $\delta^{13}\text{C}$ are mirrored by curves of major nannoplankton species at Site 690 suggesting the possibility that a significant nannoplankton community turnover was responsible to the $\delta^{13}\text{C}$ “wiggle” within the CIE onset³⁶. However, the origin of the wiggle as a result of nanno assemblage changes was pretty speculative at Site 690 and so far there is no evidence of this correlation elsewhere.

4. Presentation of the Wright / Schaller / Miller ideas.

Wright and Schaller (2014) thought they saw bedding couplets throughout the Marlboro Clay at Millville which they proposed were annual and provided a direct chronology. Myself and Ellen Thomas (2015) in *Climate of the Past* (not cited here) showed evidence from the cores themselves (concentric grooves, side injection of mud, microfaulting, etc) that the supposed couplets were a common sort of drilling artefact and therefore not a bedding feature at all. If the authors disagree with that evidence they should say why. Otherwise, line 44 should not start the discussion by stating that there are in fact bedding couplets in the Marlboro and 155-156 should not refer to bedding couplets because there aren't any.

Also, regarding ‘other evidence’ that the duration of the Marlboro cannot be measured in years, the microfossil accumulation rates discussed in Pearson and Thomas are also fundamental.

Response: We revised the sentence in the second paragraph on this issue:

“At one extreme, the CIE onset was estimated to have spanned only 13 years based on assumed annual “bedding” couplets at a paleo-shelf section on the mid-Atlantic Coastal Plain¹⁹, an assumption contradicted by evidence for coring artefacts produced via biscuiting whereby the formation is fractured during coring and drilling mud is injected in between layers. The 13-year duration is also contradicted by evidence from foraminifer accumulation rates^{37, 38}, and carbon cycle/climate modeling^{24, 39}.”

And, as a matter of fact, does Howard's Tract show anything resembling couplets, whether sedimentary or drilling artefact?

Response: We added one sentence in the Method:

“Unlike cores at Millville and Wilson Lake¹⁹, Howards Tract cores show no evidence of couplets.”

In summary I think all these questions are answerable and the issues can be rectified which should help the presentation.

Response: We appreciate all these questions, which are very helpful for the improvement of this manuscript.

References cited in the response:

1. Zeebe RE, Lourens LJ. Solar System chaos and the Paleocene–Eocene boundary age constrained by geology and astronomy. *Science* **365**, 926-929 (2019).
2. Lourens LJ, *et al.* Astronomical pacing of late Palaeocene to early Eocene global warming events. *Nature* **435**, 1083-1087 (2005).
3. Li M, Kump LR, Hinnov LA, Mann ME. Tracking variable sedimentation rates and astronomical forcing in Phanerozoic paleoclimate proxy series with evolutionary correlation coefficients and hypothesis testing. *Earth and Planetary Science Letters* **501**, 165-179 (2018).
4. Kirtland Turner S, Sexton PF, Charles CD, Norris RD. Persistence of carbon release events through the peak of early Eocene global warmth. *Nature Geoscience* **7**, 748-751 (2014).
5. Zachos JC, McCarren H, Murphy B, Röhl U, Westerhold T. Tempo and scale of late Paleocene and early Eocene carbon isotope cycles: Implications for the origin of hyperthermals. *Earth and Planetary Science Letters* **299**, 242-249 (2010).
6. Galeotti S, *et al.* Orbital chronology of Early Eocene hyperthermals from the Contessa Road section, central Italy. *Earth and Planetary Science Letters* **290**, 192-200 (2010).

7. Lauretano V, Zachos JC, Lourens LJ. Orbitally Paced Carbon and Deep - Sea Temperature Changes at the Peak of the Early Eocene Climatic Optimum. *Paleoceanography and Paleoclimatology* **33**, 1050-1065 (2018).
8. Lunt DJ, Ridgwell A, Sluijs A, Zachos J, Hunter S, Haywood A. A model for orbital pacing of methane hydrate destabilization during the Palaeogene. *Nature Geosci* **4**, 775-778 (2011).
9. Sexton PF, *et al.* Eocene global warming events driven by ventilation of oceanic dissolved organic carbon. *Nature* **471**, 349 (2011).
10. Kodama KP, Hinnov L. *Rock Magnetic Cyclostratigraphy*. Wiley-Blackwell (2015).
11. Killick R, Eckley I. changepoint: An R package for changepoint analysis. *Journal of statistical software* **58**, 1-19 (2014).
12. Bralower TJ, *et al.* Evidence for Shelf Acidification during the Onset of the Paleocene-Eocene Thermal Maximum. *Paleoceanography and Paleoclimatology* **33**, 1408-1426 (2018).
13. Aziz HA, *et al.* Astronomical climate control on paleosol stacking patterns in the upper Paleocene-lower Eocene Willwood Formation, Bighorn Basin, Wyoming. *Geology* **36**, 531-534 (2008).
14. Le Quéré C, *et al.* Trends in the sources and sinks of carbon dioxide. *Nature Geoscience* **2**, 831-836 (2009).
15. Penman DE, *et al.* An abyssal carbonate compensation depth overshoot in the aftermath of the Palaeocene-Eocene Thermal Maximum. *Nature Geosci* **9**, 575-580 (2016).
16. Li M, *et al.* Paleoclimate proxies for cyclostratigraphy: Comparative analysis using a Lower Triassic marine section in South China. *Earth-Sci Rev* **189**, 125-146 (2019).
17. Podrecca LG, Makarova M, Miller KG, Browning JV, Wright JD. Clear as mud: Clinoform progradation and expanded records of the Paleocene-Eocene Thermal Maximum. *Geology* **49**, 1441-1445 (2021).
18. Makarova M, Wright JD, Miller KG, Babila TL, Rosenthal Y, Park JI. Hydrographic and ecologic implications of foraminiferal stable isotopic response across the U.S. mid-Atlantic continental shelf during the Paleocene-Eocene Thermal Maximum. *Paleoceanography* **32**, 56-73 (2017).
19. Wright JD, Schaller MF. Evidence for a rapid release of carbon at the Paleocene-Eocene thermal maximum. *Proceedings of the National Academy of Sciences* **110**, 15908-15913 (2013).
20. Kopp RE, *et al.* An Appalachian Amazon? Magnetofossil evidence for the development of a tropical river-like system in the mid-Atlantic United States during the Paleocene-Eocene thermal maximum. *Paleoceanography* **24**, n/a-n/a (2009).
21. Zhu Z, Aller RC, Mak J. Stable carbon isotope cycling in mobile coastal muds of Amapá, Brazil. *Continental Shelf Research* **22**, 2065-2079 (2002).
22. Aller RC, Hannides A, Heilbrun C, Panzeca C. Coupling of early diagenetic processes and sedimentary dynamics in tropical shelf environments: the Gulf of Papua deltaic complex. *Continental Shelf Research* **24**, 2455-2486 (2004).
23. Self-Trail JM, *et al.* Shallow marine response to global climate change during the Paleocene-Eocene Thermal Maximum, Salisbury Embayment, USA. *Paleoceanography* **32**, (2017).
24. Zeebe RE, Ridgwell A, Zachos JC. Anthropogenic carbon release rate unprecedented during the past 66 million years. *Nature Geosci* **9**, 325-329 (2016).

25. Self-Trail JM, Powars DS, Watkins DK, Wandless GA. Calcareous nannofossil assemblage changes across the Paleocene–Eocene Thermal Maximum: Evidence from a shelf setting. *Marine Micropaleontology* **92**, 61-80 (2012).
26. Hollis CJ, *et al.* The Paleocene–Eocene Thermal Maximum at DSDP Site 277, Campbell Plateau, southern Pacific Ocean. *Clim Past* **11**, 1009-1025 (2015).
27. Bains S, Corfield RM, Norris RD. Mechanisms of Climate Warming at the End of the Paleocene. *Science* **285**, 724-727 (1999).
28. Jiang S, Wise SW. Distinguishing the influence of diagenesis on the paleoecological reconstruction of nannoplankton across the Paleocene/Eocene Thermal Maximum: An example from the Kerguelen Plateau, southern Indian Ocean. *Marine Micropaleontology* **72**, 49-59 (2009).
29. Zachos JC, *et al.* Rapid acidification of the ocean during the Paleocene-Eocene thermal maximum. *Science* **308**, 1611-1615 (2005).
30. Zachos JC, *et al.* Extreme warming of mid-latitude coastal ocean during the Paleocene-Eocene Thermal Maximum: Inferences from TEX86 and isotope data. *Geology* **34**, 737-740 (2006).
31. Sluijs A, *et al.* Environmental precursors to rapid light carbon injection at the Palaeocene/Eocene boundary. *Nature* **450**, 1218-1221 (2007).
32. Dupuis C, *et al.* The Dababiya Quarry Section: Lithostratigraphy, clay mineralogy, geochemistry and paleontology. *Micropaleontology* **49**, 41-59 (2003).
33. Sluijs A, Bijl PK, Schouten S, Röhl U, Reichert G-J, Brinkhuis H. Southern ocean warming, sea level and hydrological change during the Paleocene-Eocene thermal maximum. *Climate of the Past* **7**, 47-61 (2011).
34. Röhl U, Bralower T, Norris R, Wefer G. New chronology for the late Paleocene thermal maximum and its environmental implications. *Geology* **28**, 927-930 (2000).
35. Thomas DJ, Zachos JC, Bralower TJ, Thomas E, Bohaty S. Warming the fuel for the fire: Evidence for the thermal dissociation of methane hydrate during the Paleocene-Eocene thermal maximum. *Geology* **30**, 1067-1070 (2002).
36. Bralower TJ. Evidence of surface water oligotrophy during the Paleocene-Eocene thermal maximum: Nannofossil assemblage data from Ocean Drilling Program Site 690, Maud Rise, Weddell Sea. *Paleoceanography* **17**, 13-11-13-12 (2002).
37. Pearson PN, Nicholas CJ. Layering in the Paleocene/Eocene boundary of the Millville core is drilling disturbance. *Proceedings of the National Academy of Sciences* **111**, E1064-E1065 (2014).
38. Pearson PN, Thomas E. Drilling disturbance and constraints on the onset of the Paleocene-Eocene boundary carbon isotope excursion in New Jersey. *Climate of the Past* **11**, 95-104 (2015).
39. Zeebe RE, Dickens GR, Ridgwell A, Sluijs A, Thomas E. Onset of carbon isotope excursion at the Paleocene-Eocene thermal maximum took millennia, not 13 years. *Proceedings of the National Academy of Sciences* **111**, E1062-E1063 (2014).

REVIEWER COMMENTS

Reviewer #1 (Remarks to the Author):

This review is a re-review of a greatly revised manuscript. I have read the marked up manuscript. I read the unmarked manuscript. I read the response to the reviewers.

This is an important paper that should be published. It presents two robust datasets (MS and CaCO₃ concentrations) from Howard's Tract in Maryland that present cyclostratigraphy through the onset and duration of most of the PETM from paleoshelf marine sediments. The authors have responded adequately to all my comments from my first review. I couldn't find anything substantial to criticize or comment about in the revised manuscript.

KPK August 4, 2022

Reviewer #2 (Remarks to the Author):

Overall I found the revised version of the manuscript a significant improvement and some of the issues raised in my review have been handled adequately. The discussion of siderite diagenesis in particular is very welcome and clear. Also the definition of the onset and the discussion of isotope change points is now much clearer.

It is much better to see the carbon isotope data plotted as points but the question relating to carbon isotope sampling was not answered adequately. What I want to know is were samples taken and run for bulk stable isotopes but which failed to produce reliable data, as I suspect happened from the regular sampling pattern with data gaps, as explained in the review)? - and if so what criteria were used to reject data points? It is a minor issue and not something to accept or reject a paper by, but please explain the data processing better. The low carbonate interval and drop out of analytically measurable d13C is something I'd like to understand better but is a bit opaque still as presented.

I thank the authors for producing the core photographs in the Supplemental Info as requested, albeit at quite low resolution – if higher resolution photos are available that would be better. Unfortunately these photographs do not reassure me as regards the supposedly gradational base to the Marlboro Clay Formation and raise additional questions and uncertainty in my mind. There are two inter-related issues, one a matter of observation and the other of procedure and nomenclature, – 1) is the lithology gradational without sharp boundaries that might betray a hiatus or marked sedimentation rate change such as would affect the chronology?, and 2) is the base of the Marlboro Clay Fm placed at a lithologic change (as is normal by stratigraphic convention, e.g. N. American Stratigraphic Guide, Int. Stratigraphic Guide) rather than an invisible isotope change (simplistically, not allowed)?

The first question is the most material because if there is a hiatus, as others have described at the Aquia Fm-Marlboro Fm contact, then we would not have the full duration of the CIE onset recorded in these cores. The 6 kyr conclusion seems to depend on continuity and indeed quasi-continuous sediment deposition rate that can reasonably be tuned to a cyclic orbital solution.

The core photos are Supplementary Figs 12-14. The critical bit of the core is in Box 19 of Fig 13. I cannot reproduce it as a snip in the review because only text is allowed.

I understand that the core has desiccated and this can involve substantial colour change, and that the white blobs are where water has been spattered on the core and not important. However I cannot agree that the image supports the revised text interpretation, viz

“The contact between the Aquia Formation and the Marlboro Clay is gradational with decreasing coarse fraction and CaCO₃ content, and a gradual color change from dark greenish gray to brownish gray.”

“Observation of the HT cores suggests the contact between the Aquia Formation and the overlying

Marlboro Clay is very gradational. The top Aquia Formation is greenish-black, laminated sandy clay, while the overlying Marlboro Clay is laminated and silty clay with a color change gradually from dark greenish gray to brownish gray. Therefore, this is no evidence of a disconformity at the base of the Marlboro Clay at HT12”

Please see the annotated snip of the photo for what follows:

Instead what I see is a sharp transition about 1 inch of core below where the big red arrow is. I see nothing this abrupt in the rest of the core photos. I also wonder what the dark blobs below and above the red arrow represent – they do not look like burrows up-piping sediment but what are they? , and what is the dark bit of core beside the ‘lbo’ of Marlboro. Are these features sedimentary or diagenetic? In particular was this interval sampled for isotopes and could it be responsible for the ‘wiggle’ reversal in the carbon isotope record queried in the review?

Regarding issue 2), that of nomenclature, it seems very odd that a lithostratigraphic formation boundary is placed somewhere where nothing can be seen in the core, but so close to the obvious change. Is the arrow in the right place?

The fact – as I see it – that there is a sharp lithologic boundary at the critical point in the record is not necessarily fatal to the general argument, as one might expect such a massive environmental perturbation to be reflected in sediment even in a continuously accumulating pile. However I would expect this argument to be made in a more cogent way, with reference to the core photos, than it is at present. Or, if there might be a hiatus of indeterminate duration, how would that affect the conclusions?

Finally, in response to a query as to whether the cores show anything resembling couplets, whether sedimentary or drilling artefact, the authors say “there is no evidence of couplets”. And yet I see many such features in the core photographs, albeit at low resolution, - cm-scale light/dark bands, bending down at the edges, throughout the core. These are ostensibly similar to those previously used by some to argue for annual varve-like sedimentation. It would be interesting if the authors showed some close up photographs of these features and determine whether they appear sedimentary or artifactual.

what are these blobs?

Was this dark bit sampled? What is it?

why is the formation boundary placed here?

This contact appears sharp

The core seems to have cm scale banding throughout –

Sedimentary or drilling biscuits?

Box 18

Box 19

657.2

659.2

This is the comment.

This is the response.

This is the revised text.

REVIEWER COMMENTS

Reviewer #1 (Remarks to the Author):

This review is a re-review of a greatly revised manuscript. I have read the marked up manuscript. I read the unmarked manuscript. I read the response to the reviewers.

This is an important paper that should be published. It presents two robust datasets (MS and CaCO₃ concentrations) from Howard's Tract in Maryland that present cyclostratigraphy through the onset and duration of most of the PETM from paleoshelf marine sediments. The authors have responded adequately to all my comments from my first review. I couldn't find anything substantial to criticize or comment about in the revised manuscript.

Response: Thank you!

Reviewer #2 (Remarks to the Author):

Overall I found the revised version of the manuscript a significant improvement and some of the issues raised in my review have been handled adequately. The discussion of siderite diagenesis in particular is very welcome and clear. Also the definition of the onset and the discussion of isotope change points is now much clearer.

It is much better to see the carbon isotope data plotted as points but the question relating to carbon isotope sampling was not answered adequately. What I want to know is were samples taken and run for bulk stable isotopes but which failed to produce reliable data, as I suspect happened from the regular sampling pattern with data gaps, as explained in the review)? - and if so what criteria were used to reject data points? It is a minor issue and not something to accept or reject a paper by, but please explain the data processing better. The low carbonate interval and drop out of analytically measurable $\delta^{13}\text{C}$ is something I'd like to understand better but is a bit opaque still as presented.

Response: This question regarding isotope analyses comes down to standard lab analytical procedures. If a bulk sediment sample had sufficient carbonate content (>0.5%) to get a detectable signal, we measure it. Analytically, the mass limit for our instrument is set by gas pressure/beam size, balancing of sample vs ref gas ($\pm 1\%$), the same criteria for all analyses, NOT by the isotope values. We reported all the data for samples that met the minimum mass balancing criteria. With bulk sediment on the Kiel device we typically will add as much powdered sample as possible to generate sufficient CO₂ to meet the balancing cutoff, up to 0.5 mg.

I thank the authors for producing the core photographs in the Supplemental Info as requested, albeit at quite low resolution – if higher resolution photos are available that would be better.

Unfortunately these photographs do not reassure me as regards the supposedly gradational base to the Marlboro Clay Formation and raise additional questions and uncertainty in my mind. There are two inter-related issues, one a matter of observation and the other of procedure and nomenclature, –

1) is the lithology gradational without sharp boundaries that might betray a hiatus or marked sedimentation rate change such as would affect the chronology?, and

Response: While there is quite often a break at the Aquia/Marlboro contact, I (Jean M. Self-Trail) specifically noted on the litho log at the drill site that the "contact with the Aquia below is very gradational and gorgeous". There is a color change and an increase in sand/glaucinite, but there is no obvious break. Additionally, there isn't a big changeover in the nannoflora that would suggest missing time. I've looked at a LOT of Aquia/Marlboro contacts and would say this one is probably the most complete I've ever seen.

2) is the base of the Marlboro Clay Fm placed at a lithologic change (as is normal by stratigraphic convention, e.g. N. American Stratigraphic Guide, Int. Stratigraphic Guide) rather than an invisible isotope change (simplistically, not allowed)?

Response: Yes--the sediments change from a silty clay above (Dark greenish gray 5g4/1; Marlboro Clay) to a glauconitic sandy clay below (Greenish black 5gy 2/1; Aquia Formation). This is consistent with the definitions of both formations and follows the Code.

The first question is the most material because if there is a hiatus, as others have described at the Aquia Fm-Marlboro Fm contact, then we would not have the full duration of the CIE onset recorded in these cores. The 6 kyr conclusion seems to depend on continuity and indeed quasi-continuous sediment deposition rate that can reasonably be tuned to a cyclic orbital solution.

The core photos are Supplementary Figs 12-14. The critical bit of the core is in Box 19 of Fig 13. I cannot reproduce it as a snip in the review because only text is allowed.

I understand that the core has desiccated and this can involve substantial colour change, and that the white blobs are where water has been spattered on the core and not important. However I cannot agree that the image supports the revised text interpretation, viz

“The contact between the Aquia Formation and the Marlboro Clay is gradational with decreasing coarse fraction and CaCO₃ content, and a gradual color change from dark greenish gray to brownish gray.”

“Observation of the HT cores suggests the contact between the Aquia Formation and the overlying Marlboro Clay is very gradational. The top Aquia Formation is greenish-black, laminated sandy clay, while the overlying Marlboro Clay is laminated and silty clay with a color change gradually from dark greenish gray to brownish gray. Therefore, this is no evidence of a disconformity at the base of the Marlboro Clay at HT12”

Please see the annotated snip of the photo for what follows:

Instead what I see is a sharp transition about 1 inch of core below where the big red arrow is. I see nothing this abrupt in the rest of the core photos. I also wonder what the dark blobs below and above the red arrow represent – they do not look like burrows up-piping sediment but what are they? , and what is the dark bit of core beside the ‘lbo’ of Marlboro. Are these features sedimentary or diagenetic? In particular was this interval sampled for isotopes and could it be responsible for the ‘wiggle’ reversal in the carbon isotope record queried in the review?

Response: Almost all of these concerns were derived from the low-resolution photos with alternate colors, which was actually misleading. Thanks for pushing us to search deep and provide convincing photos in this revision. The photographs provided in the previous SI were taken during the late research stage of XRF scanning. During this stage, the core photos are limited in quality due to the white blobs and fingerprints covering the surface of the cores. However, these are not original sedimentary features.

We now supplement these photos with high-resolution photos gathered during core logging when everything was fresh. The new photos are images from HT2 at the Aquia/Marlboro Clay contact and from directly above and below. Based on these new high-resolution photos, the transition between the Aquia/Marlboro Clay is gradational with a gradual color change from dark greenish gray to brownish gray.

So, “a sharp transition about 1 inch of core below where the big red arrow is” is not a sedimentary feature. The dark blobs are areas with original color, while the light gray areas are not. We plot isotope data and core photo together, the “wiggle” was not there.

(new) Supplementary Figure 14 High-resolution core photos near the Aquia/Marlboro contact shown with $\delta^{13}\text{C}$ (black dot) and $\delta^{18}\text{O}$ (red square) data. The sediments change from a silty clay above (Dark greenish gray 5g4/1; Marlboro Clay) to a glauconitic sandy clay below (Greenish black 5gy 2/1; Aquia Formation) with no obvious break. Additionally, there isn't a big changeover in the nannoflora that would suggest missing time.

Regarding issue 2), that of nomenclature, it seems very odd that a lithostratigraphic formation boundary is placed somewhere where nothing can be seen in the core, but so close to the obvious change. Is the arrow in the right place?

The fact – as I see it – that there is a sharp lithologic boundary at the critical point in the record is not necessarily fatal to the general argument, as one might expect such a massive environmental

perturbation to be reflected in sediment even in a continuously accumulating pile. However I would expect this argument to be made in a more cogent way, with reference to the core photos, than it is at present. Or, if there might be a hiatus of indeterminate duration, how would that affect the conclusions?

Response: The pick at the drill site was gradational and very near 657.7 ft. Because it was gradational, it was harder to see when the sediments were wet. Looking at the photo of core that has dried a bit, I (Jean M. Self-Trail) would probably change the contact to 657.6 ft, where you can see the lowest amount of MC lithology in the photo.

Based on our observation and high resolution core photo, we argue that there is no hiatus of indeterminate duration and thus the estimated duration of the CIE onset should not be affected. And the minor change of the Aquia/Marlboro contact doesn't change our conclusion on the timing of the PETM at HT.

Finally, in response to a query as to whether the cores show anything resembling couplets, whether sedimentary or drilling artefact, the authors say "there is no evidence of couplets". And yet I see many such features in the core photographs, albeit at low resolution, - cm-scale light/dark bands, bending down at the edges, throughout the core. These are ostensibly similar to those previously used by some to argue for annual varve-like sedimentation. It would be interesting if the authors showed some close up photographs of these features and determine whether they appear sedimentary or artifactual.

Response: Those features you see are fractures in the clay. Our best interpretation is that the clay expanded as we were drilling to form those "biscuits" (they're really shaped more like mustaches in the split core) with the center point raised. They are not sedimentary features. We know this because each peak is in the center of the core.

Figure. High resolution photo of HT2 (655.2-657.2 ft).